# Nef stabilizes actin to prevent HIV-1 sensing by RIG-I-like receptors

Alexandre Laliberté [1,7], Caterina Prelli Bozzo[1,2,7], Dhiraj Acharya[3], Aurora De Luna [1], Maximilian Hirschenberger [1], Junji Zhu [3], Meta Volcic [1], Bettina Stolp[4], Cristina M. Rodriguez-Quinteros[4], Oliver T. Fackler [4,5], Michaela U. Gack [3], Konstantin M. J. Sparrer [1,6] ✉ & Frank Kirchhoff [1] ✉

Sensing of viral pathogens by RIG-I-like receptors (RLRs) requires their priming via dephosphorylation mediated by the protein phosphatase 1 regulatory subunit 12 C (R12C), which is activated upon virus-induced actin rearrangements. Here, we show that the HIV-1 accessory protein Nef prevents R12C-mediated RLR priming, thereby suppressing viral sensing. HIV-1 variants containing single point mutations in Nef (F/R191A) that ablate its ability to bind the actin-modulating kinase PAK2 trigger increased interferon (IFN) responses in primary CD4⁺ T cells, macrophages, and dendritic cells. Neutralization of IFN suppresses innate immune activation and enhances the replication of Nef-mutated HIV-1. We further demonstrate that HIV-1 encoding Nef F/R191A is sensed by MDA5 after proviral integration in an R12C-dependent manner. Mechanistically, PAK2 binding by Nef promotes actin repair and stabilization, thereby preventing re-localization of R12C to MDA5 and RIG-I and their subsequent dephosphorylation. Our data identify Nef as an antagonist of actin-R12C-mediated RLR priming, enabling HIV-1 to escape immune control.

Effective and specific detection of invading viral pathogens is a prerequisite for immune control and elimination. To achieve this, humans and other species have evolved a variety of immune sensors, called pattern recognition receptors (PRRs), that recognize pathogen-associated molecular patterns (PAMPs) present during infections[1–3]. Important PRRs include Toll-like receptors (TLRs) that detect various extracellular or endosomal/lysosomal PAMPs and signal via specific adaptors, such as myeloid differentiation primary response 88 (MyD88). Another PRR, the DNA sensor cGAS (cyclic GMP-AMP synthase), produces the second messenger cGAMP upon detection of viral dsDNA in the cytoplasm[4,5], which subsequently induces signaling via the adaptor STING (stimulator of interferon genes). Cytoplasmic RNAs are predominantly sensed by RIG-I-like receptors (RLRs), such as RIG-I and MDA5, that signal through the adaptor molecule mitochondrial

antiviral signaling protein (MAVS)[1,6]. Activation of all these PRRs by PAMPs eventually triggers innate immune responses, including the induction of interferons (IFNs) and other (pro-)inflammatory cytokines. The binding of these cytokines to the secreting (autocrine) or neighboring (paracrine) cells induces a transcriptional program that sets the cells in an antiviral state. This includes the upregulation of antiviral effectors, also called restriction factors, that suppress viral replication[7–9].

While detection by immune sensors is paramount for effective control and elimination of viral pathogens, excessive or improperly terminated immune activation causes and/or contributes to pathogenesis. For example, induction of cytokine storms during SARS-CoV-2 infection is linked to severe outcomes of COVID-19[10,11]. Chronic viruses, such as HIV-1, are associated with persistent inflammation that drives

[1]Institute of Molecular Virology, Ulm University Medical Center, Ulm, Germany. [2]Department of Microbial Pathogenesis, Yale University School of Medicine, New Haven, CT, USA. [3]Florida Research and Innovation Center, Cleveland Clinic, Port Saint Lucie, USA. [4]Department of Infectious Diseases, Integrative Virology, CIID, Heidelberg University, Medical Faculty Heidelberg, Heidelberg, Germany. [5]German Centre for Infection Research (DZIF), Partner Site Heidelberg, Heidelberg, Germany. [6]German Center for Neurodegenerative Diseases (DZNE), Ulm, Germany. [7]These authors contributed equally: Alexandre Laliberté, Caterina Prelli Bozzo. ✉e-mail: Konstantin.Sparrer@uni-ulm.de; Frank.Kirchhoff@uni-ulm.de

disease progression[12]. Thus, premature or aberrant activation of immune sensors must be prevented. For RLRs, it was long thought that their activation is mediated solely by the recognition of RNA-PAMPs that arise during viral infections. For example, RIG-I is activated by RNA with double-stranded stretches and containing a di- or tri-phosphate moiety at their 5' end, characteristic features of viral replication intermediates. Recently, however, it has been shown that RLRs remain inactive until they are primed by dephosphorylation via protein phosphatase 1 (PP1)[13,14]. This dephosphorylation is mediated by the PP1 regulatory subunit R12C (or PPP1R12C), which in turn is activated by viral infection-induced disturbances of the actin cytoskeleton. These virus-induced rearrangements of the actin cytoskeleton network trigger relocalization of R12C from actin filaments to cytoplasmic RLRs, and are required for effective induction of antiviral immune responses[14].

To avoid effective immune control, successful viral pathogens such as HIV-1, the causative agent of AIDS, have evolved sophisticated evasion mechanisms and countermeasures[15–18]. Although a variety of PAMPs are generated during HIV-1 reverse transcription (RT), as well as by proviral transcription, HIV-1 is barely sensed by cytoplasmic PRRs, and infected cells do not mount effective innate immune responses[17,19–22]. This can be partially explained by known viral immune evasion mechanisms. For example, recent evidence shows that the HIV-1 capsid remains intact until it reaches the nucleus, shielding various RT intermediates from cytoplasmic sensors[23]. Residual sensing of viral DNAs by cGAS is attenuated by Three prime repair exonuclease 1 (TREX1)[24]. However, it remains unclear how HIV-1 avoids RLR-mediated sensing of the various RNA species produced post-integration, including intron-containing RNAs[22] and partially double-stranded RNAs, which are expected to trigger RNA-sensing PRRs.

The accessory factors of HIV-1, Vif, Vpu, Vpr, and Nef, are well-known for their ability to promote viral immune evasion, either by directly targeting antiviral factors or the signaling cascades inducing them. For example, Vpu directly counteracts tetherin, which inhibits virus release, but also manipulates NF-κB signaling[25]. The accessory proteins are multifunctional and often dispensable for HIV-1 replication in cell lines but critical for efficient viral spread and pathogenicity in infected individuals[26,27]. None of them has previously been implicated in evasion of RLR sensing. Notably, however, the HIV-1 protein Nef recruits active pools of the host p21-activated kinase 2 (PAK2) for inactivation of the actin-binding protein cofilin and thus reduction of depolymerization and cellular actin dynamics[28–30]. PAK2-mediated control of host cell actin dynamics enables Nef to impair the morphological plasticity and motility of HIV-1 infected CD4+ T cells, and provides HIV-1 with a replication advantage in vivo[29,31–37]. While the actin-modulating properties of Nef have been known for decades[28,34], their functional relevance for HIV-1 replication and immune recognition remained unclear.

Here, we show that Nef inhibits sensing of HIV-1 by RLRs and subsequent type I IFN responses in a PAK2- and R12C-dependent manner. HIV-1 variants that are unable to associate with PAK2 elicit heightened antiviral responses and are attenuated in primary human cells due to increased IFN induction. Our data further show that HIV-1 strains containing single point mutations in Nef that specifically disrupt PAK2 binding are sensed by MDA5 after proviral integration. In contrast, in the presence of wild-type (WT) Nef, MDA5 remains in an inactive phosphorylated state, preventing effective sensing. Our results suggest that Nef-mediated modulation of the actin cytoskeleton plays a key role in HIV-1 immune evasion.

## Results
### Mutations disrupting Nef's association with PAK2 enhance HIV-1 immune sensing
To assess whether Nef plays a role in evasion of HIV-1 immune sensing, we infected primary monocyte-derived macrophages (MDM) with the clade C transmitted-founder (TF) infectious molecular clone (IMC) of HIV-1 CH042[38]. Although TF HIV-1 strains are critical for initiating infection in humans, they infect primary MDM much less efficiently than lab-adapted or M-tropic HIV-1 strains[39,40]. To achieve higher infection rates, we thus used VSV-G pseudotyped HIV-1 CH042 particles. While these are not useful to study viral entry, they are well-suitable to study post-integration effects of HIV-1 in target cells since the integrated provirus is identical to that of the WT virus. Notably, it has been shown that VSV-G pseudotyped retroviral particles also induce actin disturbances[14]. In addition to WT HIV-1, we used IMCs expressing Nef with mutations in the myristylation site (G2A), the SH3 domain interaction motif (72-PxxP-75), and the PAK2 recruitment site (R191A). The G2A mutation disrupts membrane association of Nef, while mutating the PxxP motif abrogates binding to SH3 domains, including that of the guanine exchange factor Vav1, an essential component of the Nef-PAK2 complex. Mutation of the PxxP motif and R191A also abrogates the association of Nef with PAK2, and its ability to disrupt actin dynamics[41,42]. Infection by HIV-1 CH042 constructs containing the AxxA or R191A mutation in Nef was associated with ~2-fold higher mRNA expression levels of the IFN-stimulated gene (ISG) MX1 encoding myxoma resistance protein 1 (Mx1), compared to WT HIV-1 infection (Fig. 1a). In contrast, the G2A mutation did not impair Nef's ability to suppress MX1 induction. This is remarkable since the G2A change prevents myristoylation and hence membrane association of Nef, which is critical for many activities, such as downmodulation of CD4, MHC class I, and SERINC5[43,44]. In addition, it has been reported that the G2A mutation disrupts the ability of Nef to suppress LINE-1 RNA expression and consequently enhances LINE-1-triggered IFN production[45]. Since R191A disrupts PAK2 association more selectively than the mutation of the PxxP motif[41,42], we used this Nef mutant in further experiments.

Modulation of host cell actin dynamics involves hyperphosphorylation and thus inactivation of the actin-severing factor cofilin[28]. Consistently, flow cytometric analyses confirmed an increase in cofilin phosphorylation at serine 3, indicative of stabilization of actin filaments, in PBMCs infected with WT HIV-1 CH042 (Supplementary Fig. 1a, b). This effect was markedly reduced in cells infected with the R191A Nef mutant virus. Cofilin deregulation by several HIV-1 Nef variants has been linked to the impairment of actin polymerization into a circumferential F-actin ring and spreading of CD4 T cells in response to surface-mediated T cell receptor activation[28,29,34]. Testing whether cofilin deregulation by CH042 Nef also impairs CD4 + T cell actin dynamics, we found that CH042 Nef significantly reduced the fraction of cells that respond to stimulation with both actin polymerization and spreading. This effect was similar to that of the HIV-1 NL4.3 Nef control and involved the PAK2-interaction motif at position 191 for both Nef variants (Supplementary Fig. 1c, morphotype 1, white bars). Interference with host cell actin dynamics via the PAK2-cofilin axis is thus a conserved feature of NL4.3 and CH042 Nef proteins.

Analyses of a larger number of donors showed that MDMs infected with R191A Nef mutant CH042 expressed not only higher levels of MX1 but also of the proinflammatory cytokines CXCL10 (also known as IP-10) and CCL5 (RANTES) compared to cells infected with the otherwise isogenic WT virus (Fig. 1b). Notably, higher levels of immune activation by R191A Nef CH042, were not due to higher infection rates (Supplementary Fig. 2a, b). Western blot analyses revealed that only R191A Nef HIV-1, but not the WT virus, induced ISGylation in MDMs (Fig. 1c, d). Flow cytometry analysis showed that the protein levels of Mx1 also increased (Supplementary Fig. 2b, c). Control experiments showed that the Mx1 antibody used is highly specific since the signal obtained with MX1 KO cells was indistinguishable from the isotype control (Supplementary Fig. 2d, e). In further support of suppression of innate immune activation by WT HIV-1 Nef, higher levels of proinflammatory cytokines (i.e., IFNα1, IFN-γ, IFNλ1, IFNλ2/3, IL-6, IL-12, IL-10, IP-10, and IL-1β) were detected by multiplex ELISA in the

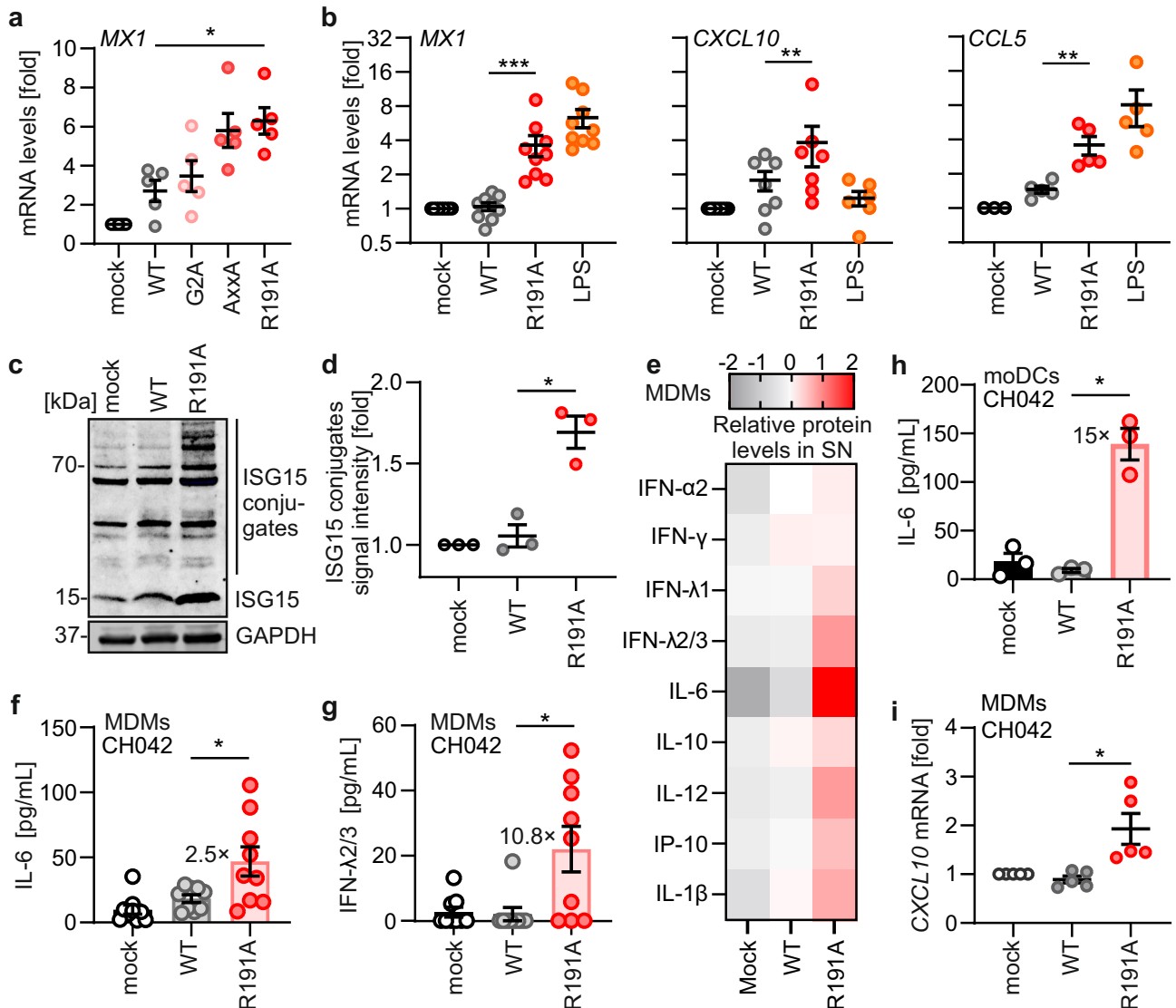

**Fig. 1 | Nef antagonizes sensing by using its functional PAK2 motif. a** qRT-PCR analysis of cellular *MX1* mRNA levels in primary MDMs infected with VSV-G pseudotyped HIV-1 CH042 expressing WT (gray) or mutant Nef (red), measured at 48 h post-infection. Data shows mean (±SEM) relative to uninfected cells (mock, white). Each point represents the mean of one donor measured in duplicates (*N* = 5 donors). **b** qRT-PCR analysis of cellular *MX1*, *CXCL10*, and *CCL5* mRNA levels in primary MDMs infected as in **a** with CH042-expressing WT Nef (gray) or Nef R191A (red) or treated with LPS (1 μg/mL; orange). Data shows mean (±SEM) relative to uninfected cells (mock). Each point represents the mean of one donor measured in duplicates (MX1, *N* = 9 donors; CXCL10, *N* = 7 donors; CCL5, *N* = 5 donors). **c** Western blot showing ISGylation (ISG15) of cellular proteins in primary MDMs infected as in (**a**) and in the presence of 0.5 mM nucleosides. **d** Quantification of total ISGylation intensity detected in (**c**) for mock (white), WT (gray), or Nef F191A (red). Data shows mean (±SEM), each point represents one donor (*N* = 3). **e** Cytokine

levels in supernatants of primary MDMs from (**b**) and analyzed by flow cytometry. Heatmap represents the average of *Z*-scores of individual donors (*N* = 9 donors). **f** and **g** Absolute concentrations of IL-6 (**f**) and IFN-λ2/λ3 (**g**) in the supernatants from (**b**). Bars represent mean (± SEM), points represent donors (*N* = 9 donors). **h** IL-6 protein levels in supernatants of monocyte-derived dendritic cells (moDCs) infected as in (**a**). Bars represent mean (±SEM), points represent donors (*N* = 3 donors). **i** qRT-PCR analysis of cellular *CXCL10* mRNA levels in primary MDMs infected with non-pseudotyped CH042 expressing either WT Nef or Nef R191A at 4dpi. Data shows mean (±SEM) relative to uninfected cells (mock; white). Each point represents the mean of one donor measured in duplicates (*N* = 4 donors). Unless otherwise indicated, statistical analysis was done using two-sided ratio paired *t* tests. *p* values are indicated as **p* < 0.05; ***p* < 0.01; ****p* < 0.001 or not significant (*p* > 0.05). Exact *P* values and Source data are provided in the Source data file.

supernatants of MDMs infected with the R191A Nef mutant virus (Fig. 1e). For example, the Nef mutant virus induced significantly higher levels of IL-6 and IFN-λ2/λ3 protein compared to WT HIV-1 (Fig. 1f, g). In line with these findings, primary human monocyte-derived dendritic cells (moDCs) responded to infection by R191A Nef HIV-1 CH042 with substantially higher levels of IL-6, IL-8, IP-10, and IL-1β compared to cells infected with WT HIV-1 (Fig. 1h; Supplementary Fig. 2f–h). To validate these findings in the context of non-pseudotyped HIV-1, we infected MDMs with genuine CH042. Both mutant and WT virus induced only modest expression of *MX1*

(Supplementary Fig. 2i). However, mutation of R191A in Nef was associated with significantly increased levels of *CXCL10*, which is directly induced by viral sensing, rather than through IFN signaling exclusively (Fig. 1i). Importantly, these effects were not due to differences in viral replication (Supplementary Fig. 2j).

Some of the effects observed were modest compared to previous studies, which counteracted SAMHD1 restriction to achieve high HIV-1 infection rates in primary cells[46,47]. To assess the effects of Nef under such experimental conditions, we infected MDMs in culture media supplemented with exogenous nucleosides. This approach increased

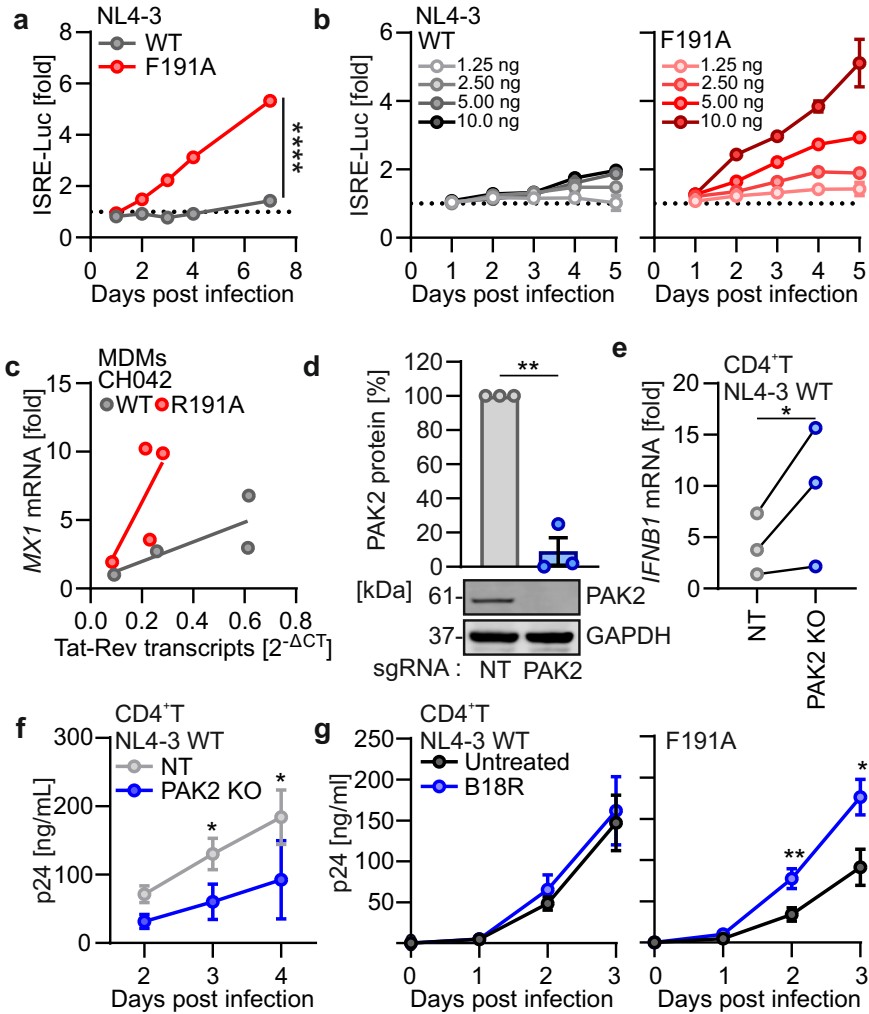

**Fig. 2 | PAK2 and actin remodeling are required for Nef-mediated antagonism of sensing. a** Relative luciferase (Luc) induction in THP-1 dual cells containing ISRE-Luciferase and NF-κB reporter genes infected with VSV-G pseudotyped NL4-3-expressing WT (gray) or F191A (red) Nef. Each point represents the mean of triplicates (±SEM). Repeated-measure two-way ANOVA, Geisser-Greenhouse's correction. **b** Relative luciferase induction in THP-1 dual reporter cells infected with the indicated amounts of p24 from VSV-G pseudotyped HIV-1 NL4-3 WT (left) or expressing Nef F191A (right). Each data point represents the mean of triplicates (± SEM). **c** Correlation between *MX1* mRNA levels measured in MDMs infected as in Fig. 1b and fully spliced HIV RNA (*tat-rev*) transcripts, normalized to *GAPDH* mRNA levels. Each point represents the mean of one donor measured in duplicates (*N* = 4 donors). Line represents linear regression. **d** Western blot (below) and quantification (above) of PAK2 in primary CD4+ T cells electroporated with Cas9/NT-sgRNA (gray) or Cas9/*PAK2*-sgRNA (blue) complex. Bars represent mean (±SEM), each point represents one donor (*N* = 3 donors). Two-sided Welch's *t* test. **e** qRT-PCR analysis of *IFNB1* mRNA in primary CD4+T cells in **d** infected with HIV-1 NL4-3 WT or Nef F191A at 4 dpi. Each point represents the value of one donor normalized to uninfected cells (*N* = 3 donors). **f** Replication kinetics of WT NL4-3 in NT (gray) control or *PAK2* KO (blue) primary CD4+ T cells. At the indicated time points, p24 levels were quantified by ELISA in the supernatants. Data shows the mean of 3 donors (±SEM). **g** Replication kinetics of WT (left) or Nef F191A mutant (right) NL4-3 in primary CD4+T cells cultured in the presence (blue) or absence (black) of the IFN-I blocker B18R. At the indicated time points, p24 levels in the supernatants were quantified by ELISA. Data shows the mean of 3 donors (±SEM). Unless otherwise indicated, statistical analysis was done using two-sided paired *t* tests. *P* values are indicated as *$p < 0.05$; **$p < 0.01$, or not significant ($p > 0.05$). Exact *P* values and Source data are provided in the Source data file.

HIV-1 infection by ~10-fold and resulted in higher levels of innate immune activation (Supplementary Fig. 3a–d). However, under these conditions, the WT Nef protein failed to significantly reduce the induction of *MX1* expression (Supplementary Fig. 3d). Further analyses revealed that addition of nucleosides increased proviral integration by ~16-fold and viral RNA transcription by ~130-fold (Supplementary Fig. 3e, f). Altogether, these results show that WT, but not R191A Nef, suppresses the induction of IFNs, proinflammatory cytokines, and ISGs in HIV-1-infected primary human cells, unless viral RNA is exceedingly high.

## PAK2 association of Nef promotes HIV-1 replication by suppressing IFN induction

Since only a fraction of the cells are infected with HIV-1 in primary cell cultures, our bulk analyses most likely underestimated the impact of

Nef on immune evasion. To address this, we infected THP-1 type I IFN reporter cells (THP-1 Dual) with p24-normalized VSV-G pseudotyped particles of the T-cell line-adapted HIV-1 NL4-3 molecular clone. The Nef F191A mutant NL4-3 virus that fails to induce cofilin hyperphosphorylation but exerts other Nef activities[28,29] induced type I IFN and NF-κB signaling by ~5-fold, while the WT virus had little if any immunostimulatory activity (Fig. 2a; Supplementary Fig. 4a). These effects were dependent on the viral dose (Fig. 2b) and also observed for the primary HIV-1 CH042 strain regardless of the differentiation status of THP-1 cells to monocytes or macrophages (Supplementary Fig. 4b, c). Increased innate immune activation is expected to restrict HIV-1 replication. Indeed, increased levels of *MX1* expression in primary human MDMs infected with the R191A Nef HIV-1 CH042 were associated with lower numbers of fully spliced viral *tat-rev* mRNA levels, a

surrogate for productive infection (Fig. 2c). To examine the involvement of PAK2 in innate immune activation and HIV-1 replication, we infected primary human CD4[+] T cells with HIV-1 NL4-3 after knock-out of *PAK2* (Fig. 2d). In *PAK2* KO cells, WT HIV-1 induced significantly higher levels of type I IFN (Fig. 2e) and showed reduced virus replication (Fig. 2f). In line with these data, PAK inhibition by FRAX597 reduced cofilin S3 phosphorylation and increased *MX1* induction by WT CH042 HIV-1 in primary MDMs (Supplementary Fig. 4d–g). Finally, we found that the F191A mutation in Nef reduces HIV-1 replication in primary CD4[+] T cells (Fig. 2g). However, the replicative fitness of the mutant virus was rescued by the IFN-I blocker B18R, indicating that impaired replication was due to heightened innate immune control (Fig. 2g). Altogether, these data show that PAK2 is required for the ability of Nef to suppress innate immune activation for efficient replication of HIV-1 in primary human cells.

## Nef antagonizes post-integration sensing of HIV-1

To dissect at which step of the viral replication cycle Nef affects sensing of HIV-1 in infected macrophages, we employed antiretroviral agents. Infection of THP-1 ISRE reporter cells with Nef R191A mutant VSV-G-pseudotyped HIV-1 NL4-3 or CH042 induced robust type I IFN responses (Fig. 3a; Supplementary Fig. 5a, b). Inhibition of viral entry by AMD3100 or Maraviroc, which block the co-receptors CXCR4 and CCR5[48], respectively, together with VSV-G neutralizing antibodies completely abrogated infection and type I IFN induction (Fig. 3a; Supplementary Fig. 5a, b). Treatment with the RT inhibitor Nevirapine or the integrase inhibitor Raltegravir (RAL) also strongly reduced p24 expression and the induction of type I IFN signaling. In contrast, the protease inhibitor Nelfinavir had only a little inhibitory effect on VSV-G-pseudotyped WT and Nef mutant HIV-1 NL4-3 and CH042 infection and the induction of type I IFN responses (Fig. 3a; Supplementary Fig. 5a, b).

To determine if proviral integration is required for actin-dependent sensing of HIV-1 in primary cells, we infected MDMs with pseudotyped HIV-1 CH042 WT or Nef R191A. Treatment with Maraviroc and VSV-G neutralizing antibody strongly reduced the expression of HIV-1 transcripts and *MX1* induction (Supplementary Fig. 5c). Similarly, treatment with RAL inhibited viral RNA expression (Supplementary Fig. 5d) and significantly reduced *MX1* mRNA induction (Fig. 3b), as well as IP-10 cytokine release (Fig. 3c). Finally, RAL also prevented HIV-1 CH042 transcription (Supplementary Fig. 5e), *MX1* induction (Fig. 3d), and IP-10 production (Fig. 3e) in moDCs. In summary, these results show that Nef mainly suppresses innate immune sensing of HIV-1 post-integration products.

## Nef antagonizes RLR-mediated HIV-1 sensing

To examine which cytoplasmic PRR(s) are responsible for increased induction of type I IFN responses by HIV-1 Nef F191A, we used *cGAS*, *MAVS*, *MDA5*, and *RIG-I* knockout THP-1 cells (Supplementary Fig. 6a, b). In WT THP-1 IFN reporter cells, HIV-1 Nef F191A infection induces a robust type I IFN signal, as measured by luciferase expression (Supplementary Fig. 6c). *cGAS* KO only moderately reduced ISRE promoter activation by HIV-1 F191A Nef infection, while type I IFN signaling was almost completely abrogated upon *MAVS* KO (Fig. 4a). To determine which RLR mediates sensing of Nef-mutated HIV-1, we analyzed type I IFN signaling induced by HIV-1 Nef F191A in THP-1 *RIG-I* KO and *MDA5* KO cells. Whereas RIG-I deficiency did not reduce ISRE promoter activation, *MDA5* KO did (Fig. 4b). To further examine the role of RLRs in HIV-1 sensing and immune-signal activation, we tested the effects of the KOs on sensing-dependent NF-κB induction. Consistent with previous data, WT Nef blocked NF-κB signaling in the parental THP-1 dual cell line (Supplementary Fig. 6d). By contrast, KO of *MAVS* or *MDA5* significantly reduced NF-κB signaling induced by HIV-1 Nef F191A, while KO of *cGAS* or *RIG-I* had no significant effect (Fig. 4c).

To further verify that RLR signaling is counteracted by Nef, we knocked out the signaling adaptor MAVS[49,50], which is crucial for RLR signaling, in primary MDMs using Cas9/gRNA complexes (Fig. 4d). Consistent with our previous data, infection with the R191A-Nef HIV-1 CH042 induced substantially higher levels of *MX1* expression than the WT virus (Fig. 4e). *MAVS* KO strongly reduced *MX1* mRNA upregulation by CH042 expressing R191A Nef but did not further affect the already poor induction by WT HIV-1 (Fig. 4e). Similarly, KO of *MDA5* in primary MDMs strongly reduced *MX1* mRNA upregulation by CH042 expressing R191A Nef (Fig. 4f, g).

ISG15 conjugation is essential for effective antiviral IFN responses mediated by MDA5[51,52]. ISGylation of MDA5 promotes its oligomerization and triggers activation of innate immunity against various viral pathogens. Overexpression of WT but not F191A Nef, reduced the ISGylation of FLAG-MDA5 in HEK293T treated with poly(I:C), indicating that Nef interferes with MDA5 activation (Fig. 4h). In primary cells infected with HIV-1, we found that MDA5 was conjugated to ISG15 only in MDMs infected with Nef mutant HIV-1, as well as in cells treated with poly(I:C). In contrast, the ISG MDA5 and ISG15ylation on MDA5 was undetectable in uninfected or WT HIV-1-infected macrophages (Fig. 4i). Altogether, these results show that Nef allows HIV-1 to avoid sensing by MDA5.

## Nef prevents R12C-mediated priming of RLRs

To clarify whether Nef antagonizes RLR sensing in the absence of other viral proteins, we stimulated Chinese hamster ovary (CHO) cells expressing doxycycline-inducible WT, G2A, F195A or P72A/P75A(AxxA) SF2 Nef with the well-known RIG-I-stimulatory agent Sendai Virus (SeV)[53]. Western blot analysis confirmed Nef expression upon doxycycline treatment (Supplementary Fig. 7a). Our results showed that expression of WT and G2A, but not F191A or AxxA Nef, effectively inhibited SeV-induced *MX1* expression (Fig. 5a). Sensing of actin disturbances involves relocalization of R12C to RIG-I and/or MDA5 to mediate their dephosphorylation and priming[14]. To clarify at which step Nef counteracts RLR activation, we transfected parental or clonal *R12C* KO HEK293T cells with proviral HIV-1 NL4-3 constructs expressing WT or F191A Nef, or lacking Nef expression entirely because of two early stop codons (*nef***). Transfection of proviral HIV-1 constructs bypasses the early steps but mimics viral RNA and protein expression by integrated proviruses. In addition, transfection also causes disturbances of the actin cytoskeleton similar to virus infection[14]. HEK293T cells lack functional DNA sensing[54]; however, HEK293T cells are functional for RLR sensing, and MDA5 and RIG-I can be activated in these cells by high-molecular-weight poly (I:C) transfection, or SeV infection, respectively. As expected, both stimuli induced *IFNB1* expression (Supplementary Fig. 7b). In contrast, transfection with the synthetic DNA G3-YSD or treatment with cGAMP, which activate cGAS and STING, respectively, did not lead to any response in these cells (Supplementary Fig. 7b). Similarly, transfections with empty control or LTR containing vectors, did not induce *IFNB1* (Supplementary Fig. 7c). Thus, HEK293T cells allow to study RLR sensing of HIV-1 RNA transcripts upon transfection of proviral constructs without interference by cGAS-STING signaling.

We found that the lack of Nef or mutation of F191A increased *IFNB1* induction by about 3- to 5-fold in WT HEK293T cells (Fig. 5b). Strikingly, induction of cellular *IFNB1* mRNA expression was completely abolished in the absence of R12C (Fig. 5b, Supplementary Fig. 7d–f). Confocal microscopy analysis revealed that R191A Nef, but not WT Nef, significantly reduced the colocalization of R12C with the actin cytoskeleton in primary MDMs infected by VSV-G pseudotyped HIV-1 CH042 (Fig. 5c). This suggests that WT Nef counteracts the release of R12C from actin filaments and its subsequent translocation to RLRs. Of note, R12C adopts a diffuse cytoplasmic distribution in the presence of HIV-1 Nef R191A. In line with this, proximity ligation assays (PLAs) demonstrated effective recruitment of endogenous R12C to

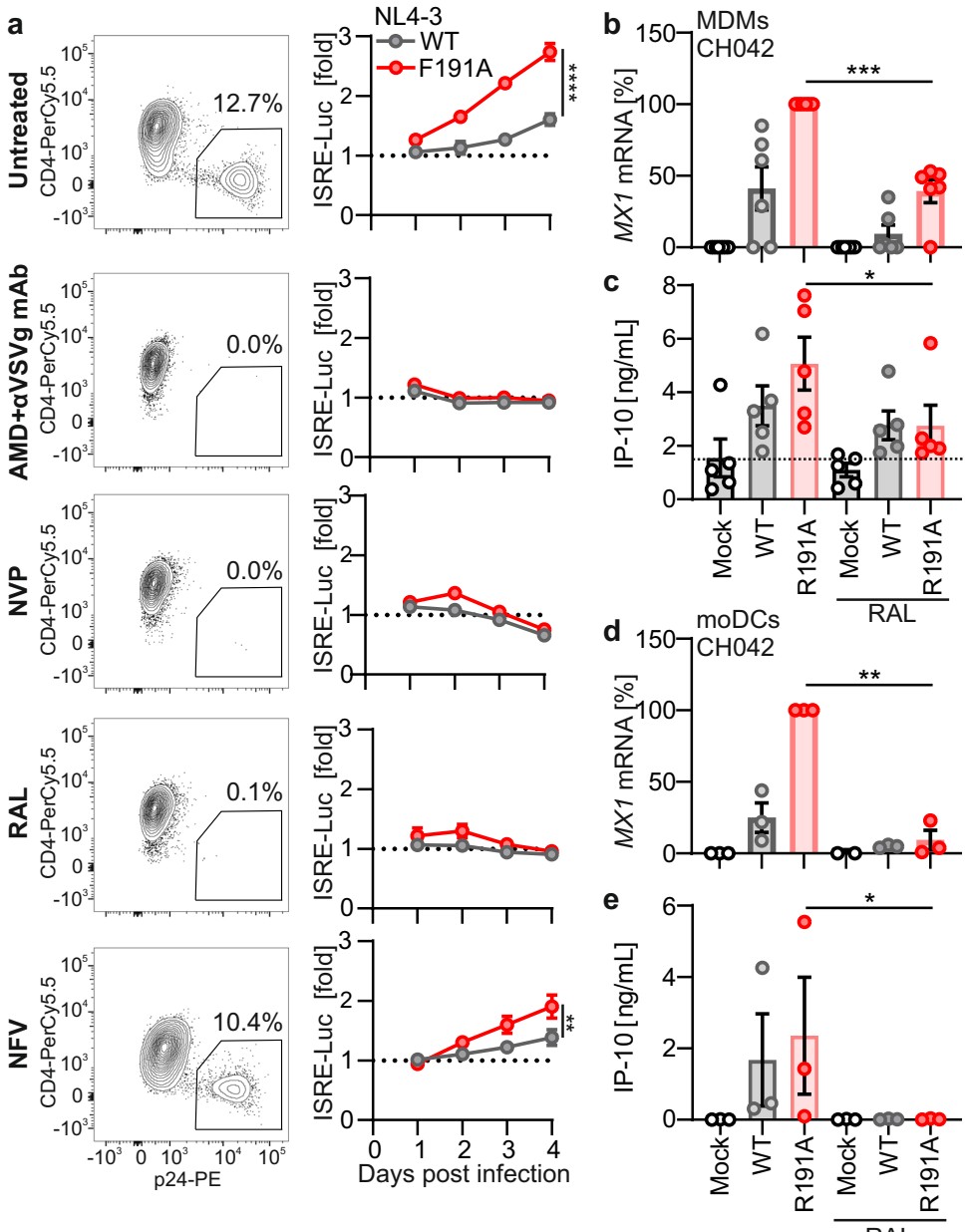

**Fig. 3 | Nef antagonizes post-integration sensing of HIV-1. a** Representative dot plot at 2 dpi showing infection levels (p24⁺ and CD4⁻) in THP-1 dual reporter cells infected with VSV-G pseudotyped HIV-1 NL4-3 WT or expressing Nef F191A and treated with the indicated drugs (left). Relative luciferase induction in THP-1 dual reporter cells infected as indicated (right; WT, gray; F191A, red). Each point represents the mean of triplicates (±SEM). Cells were either left untreated, treated with AMD3100 and anti-VSV-G neutralizing antibodies, Nevirapine (NVP), Raltegravir (RAL) or Nelfinavir (NFV) as indicated. Repeated-measure two-way ANOVA, Geisser-Greenhouse's correction. **b** qRT-PCR analysis of *MX1* mRNA induction in primary MDMs infected with VSV-G pseudotyped HIV-1 CH042 WT (gray) or Nef R191A (red) at 48 hours post-infection in the presence of Raltegravir (RAL). Values

relative to R191A-infected, untreated cells. Bars represent mean (±SEM), points represent donors (*N* = 6 donors). Two-sided Welch's *t*-test. **c** IP-10 levels in supernatants of primary MDMs infected as in (**b**). Bars represent mean (±SEM), points represent donors (*N* = 5 donors). **d** qRT-PCR analysis of *MX1* mRNA induction in primary moDCs infected as in (**b**). Values relative to R191A-infected, untreated cells. Bars represent mean (±SEM), points represent donors (*N* = 3 donors). Two-sided Welch's *t*-test. **e** IP-10 levels in supernatants of moDCs infected as in (**b**). Bars represent mean (±SEM), points represent donors (*N* = 3 donors). Unless otherwise indicated, statistical analysis was done two-sided ratio paired *t*-test. *P* values are indicated as *$p < 0.05$; **$p < 0.01$; ***$p < 0.001$, ****$p < 0.001$ or not significant ($p > 0.05$). Exact *P* values and Source data are provided in the Source data file.

MDA5 in MDMs infected with the CH042 Nef R191A mutant but not the WT virus (Fig. 5d, Supplementary Fig. 8a). Silencing of R12C by siRNA prior to infection almost entirely abolished the PLA signal, indicating that it is highly specific (Supplementary Fig. 8b-c). Since RIG-I may partially compensate for the lack of functional MDA5[55,56], we tested whether Nef also prevents relocalization of R12C to RIG-I. As observed for MDA5, R12C was recruited to RIG-I in MDMs infected with the CH042 Nef R191A mutant but not the WT virus (Supplementary

Fig. 8d). Upon its dissociation from filamentous actin, R12C forms a complex with PP1α or PP1γ; the PP1-R12C phosphatase complex is then recruited to MDA5 and triggers its dephosphorylation at S88, leading to active MDA5 downstream signaling. Purification of endogenous MDA5 from THP-1 cells infected with WT or mutant HIV-1 NL4-3 revealed that the PAK2-binding-deficient Nef F191A mutant, or Nef-defective virus, strongly reduced the levels of p-S88 phosphorylated MDA5, indicative of MDA5 activation (Fig. 5e). In contrast, WT HIV-1 did

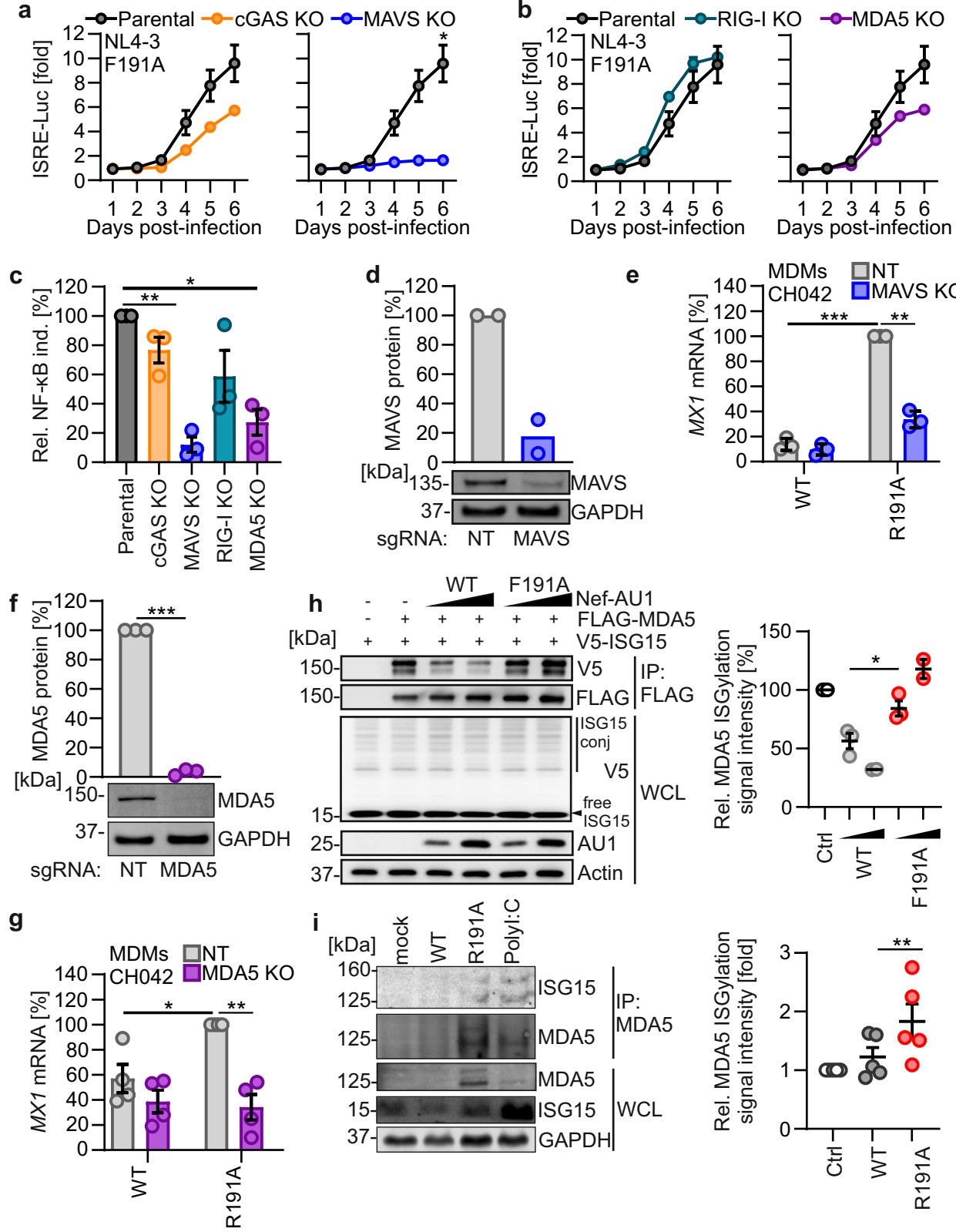

not elicit MDA5 S88 dephosphorylation (Fig. 5e). Importantly, these experiments were performed without VSV-G pseudotyping using genuine HIV-1 infection. Altogether, these results demonstrate that the HIV-1 Nef protein prevents re-localization of R12C from actin filaments to RLRs, including MDA5, to suppress their priming and subsequent innate immune activation (Supplementary Fig. 9).

## Discussion

HIV-1 is known to be a stealth master, avoiding detection by the immune system[57,58]. The underlying molecular mechanisms are, however, still incompletely understood[59–61]. Here, we show that the accessory protein Nef, by stabilizing the actin cytoskeleton, inhibits the actin-dependent activation of the PP1-regulatory protein R12C

**Fig. 4 | Nef antagonizes HIV-1 sensing by MDA5. a** and **b** Relative luciferase (LUC) induction by infection with VSV-G pseudotyped NL4-3 expressing F191A Nef in parental (gray) THP-1 dual cells, *CGAS* KO (orange), or *MAVS* KO (blue) (**a**) and parental (gray), *RIG-I* KO (petrol), or *MDA5* KO (purple) (**b**). Each point represents the mean of triplicates (±SEM). Repeated-measure two-way ANOVA, Geisser–Greenhouse's correction. **c** Relative NF-κB (SEAP) induction by infection with VSV-G pseudotyped NL4-3 expressing F191A Nef in the indicated THP-1 dual cells at 4 dpi. Points represent individual experiments, normalized to values obtained in the parental cells (*N* = 3). **d** Quantification (above) and western blot (below) of MAVS in MDMs electroporated with Cas9/NT-sgRNA (gray) or Cas9/ *MAVS*-sgRNA (blue) complex (*N* = 2 donors). **e** qRT-PCR analysis of *MX1* mRNA induction of cells in (**d**) infected with VSV-G pseudotyped HIV-1 CH042 WT or Nef R191A at 48 h post-infection (*N* = 3 donors), normalized to NT-electroporated cells infected with HIV-1 CH042 Nef R191A. **f** Quantification (above) and western blot (below) of MDA5 in MDMs electroporated with Cas9/NT-sgRNA (gray) or Cas9/

*MDA5*-sgRNA (purple) complex (*N* = 3 donors). **g** qRT-PCR analysis of *MX1* mRNA induction of cells in (**f**) infected as in (**e**) (*N* = 4 donors). **h** Western blot (left) and quantification (right) showing ISGylation (V5-ISG15) of MDA5 in HEK293T cells overexpressing ISG15-V5, MDA5-FLAG, and the indicated Nef-AU1 allele (WT, gray; F191A, red), and treated with poly (I:C) HWM-Lyovec for 12 h. Data shows mean (±SEM), each point represents one experiment, normalized to values in the absence of Nef (*N* = 2-3 independent experiments). **i** Western blot (left) and quantification (right) of ISGylation (ISG15) conjugates in immunoprecipitated MDA5 from MDMs infected as in (**e**), or treated with poly (I:C) HWM-Lyovec and in the presence of 0.5 mM nucleosides. Values are normalized to those obtained in mock cells. Data shows mean (±SEM; *N* = 5 donors). Two-sided ratio paired *t*-test. Unless otherwise indicated, bars represent mean (±SEM), points represent individual donors, and statistical analysis was done using two-sided Welch's *t* tests. *P* values are indicated as *$p < 0.05$; **$p < 0.01$; ***$p < 0.001$ or not significant ($p > 0.05$). Exact *P* values and Source data are provided in the Source data file.

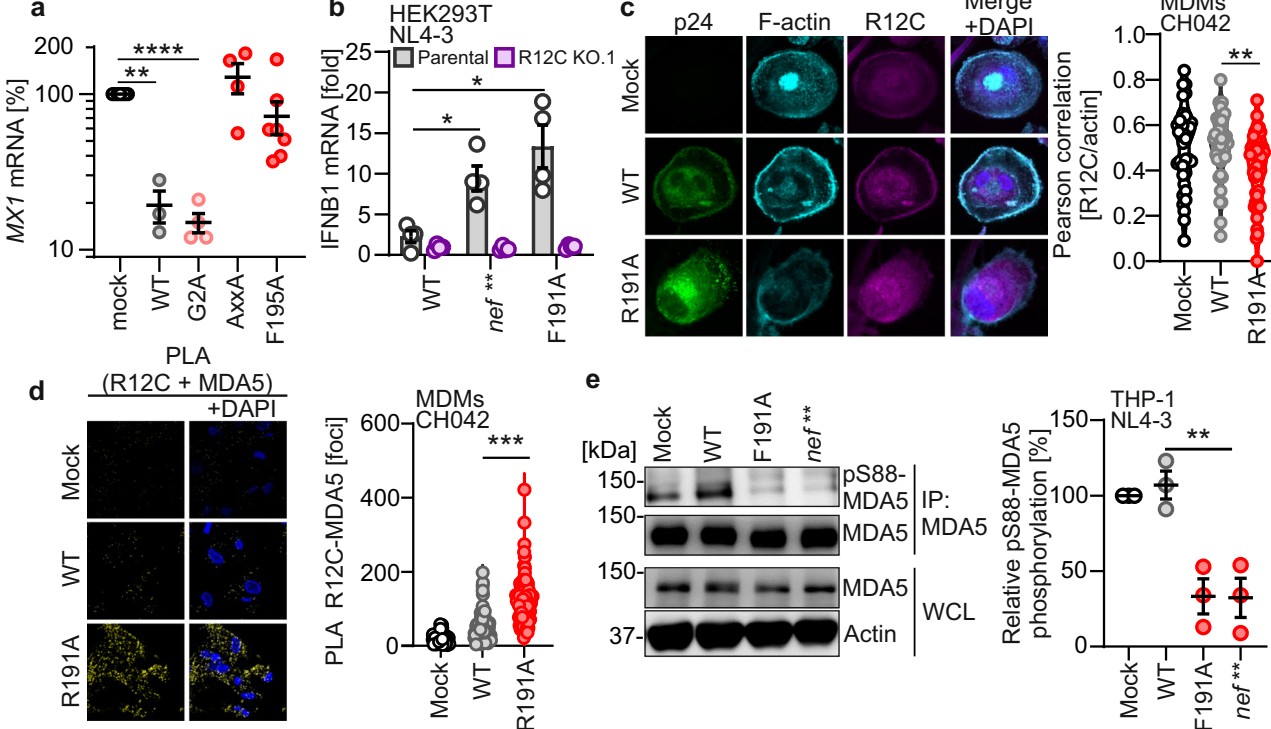

**Fig. 5 | Nef prevents R12C-mediated dephosphorylation of RLRs. a** Relative induction of the *Cricetulus griseus mx1-like* gene in CHO cells expressing the indicated Nef mutants as a doxycycline-inducible gene. CHO cells were treated with doxycycline and infected with 25 HAU SeV. *Mx1-like* gene induction was measured by qRT-PCR at 24 hpi. Values are normalized to those obtained in the absence of doxycycline. Each point represents an individual experiment (*N* = 3, WT, gray; *N* = 4, G2A, pink; *N* = 4, AxxA, red; *N* = 7, F195A, red), line indicates mean (±SEM). **b** qRT-PCR of cellular *IFNB1* mRNA levels in parental (gray) or *R12C* KO.1 (purple) HEK293T cells transfected with proviral HIV-1 NL4-3 expressing WT Nef, Nef F191A mutation or lacking Nef. Bars show mean (±SEM) relative to cells transfected with an empty vector. Each point represents an individual experiment (*N* = 5). **c** Representative confocal microscopy images (left) showing intracellular p24 (green), F-actin (cyan), R12C (magenta), and DAPI (blue) in primary MDMs infected with VSV-G pseudotyped HIV-1 CH042 WT or expressing R191A Nef in the presence

of 0.5 mM nucleosides. Scale bar: 10 μm. Quantification of Pearson colocalization between R12C and F-actin (right). Each point represents the value obtained for one cell (*N* = 36 cells, mock; *N* = 43 cells, WT; *N* = 43 cells). **d** Proximity ligation assay (left; PLA) of endogenous R12C and MDA5 in primary MDMs infected as described in (**c**). Scale bar 10 μm. Quantification of the PLA signal (right). Each point represents the number of foci per cell (*N* = 80 cells, mock; *N* = 89 cells, WT; *N* = 80 cells). **e** Western blot (left) and quantification (right) showing phosphorylation of MDA5 at S88 after pulldown of endogenous MDA5 in THP-1 cells that were either mock-treated or infected with non-pseudotyped HIV-1 NL4-3 WT (gray) or F191A (red) Nef, or lacking Nef (red). Data shows mean ( ± SEM), each point represents one experiment (*N* = 3). Unless otherwise indicated, statistical analysis was done using two-way Welch's *t* tests. *P* values are indicated as *$p < 0.05$; **$p < 0.01$; ***$p < 0.001$; ****$p < 0.0001$, or not significant ($p > 0.05$). Exact P values and Source data are provided in the Source data file.

(Supplementary Fig. 9). This prevents R12C from interacting with and dephosphorylating MDA5, a process essential for initiating effective immune responses. Our data demonstrate that this function is disrupted by single-point mutations of R/F191F that prevent the interaction of Nef with the kinase PAK2, known to be critical for Nef-mediated stabilization of actin filaments[28,29,41,42]. Our results identify Nef as an antagonist of actin disturbance sensing that is a prerequisite for RLR-

mediated intracellular dsRNA detection and induction of potent IFN responses[14]. Lack of this Nef function increases sensing of HIV-1 by RLRs in primary CD4[+] T cells, macrophages, and dendritic cells and enhances IFN-dependent control of virus replication.

HIV-1 is known for its complex impact on the actin cytoskeleton[62]. For example, the interaction of the HIV-1 envelope protein with the CXCR4 coreceptor during viral entry has been reported to activate

cofilin to transiently depolymerize actin, allowing the viral capsid to cross the actin cortex and reach the nucleus[63,64]. In contrast, Nef recruits active PAK2, thereby promoting phosphorylation of cofilin and the interaction of Nef with the exocyst complex[29,34]. Both processes synergize to reduce F-actin turnover[28,29]. At first glance, these opposing effects during viral entry and in established infection might seem contradictory. However, it is conceivable that different conditions promote various stages of the viral replication cycle. A prominent example is CD4, the receptor of HIV-1. It is essential for HIV-1 entry but detrimental for viral release and infectivity[65,66]. Furthermore, early during infection, HIV-1 stimulates NF-κB to boost viral transcription, while later it inhibits it to suppress innate immune activation[67,68]. Similarly, HIV-1 may have differential effects on actin dynamics during its replication cycle. Initially, actin destabilization may support HIV-1 entry and penetration of viral capsids through cortical actin to reach the nucleus for proviral integration and productive infection. However, post-integration, once RLR-stimulatory viral RNA products are produced, Nef inhibits cofilin and stabilizes the actin cytoskeleton to prevent sensing.

Nef is highly versatile, and its impact on the actin cytoskeleton may serve several purposes. Previously, Nef was shown to prevent host cell actin dynamics to impair morphological plasticity and motility of infected lymphocytes[28]. It will be of interest to further examine whether Nef-mediated stabilization of the actin cytoskeleton not only suppresses immune sensing and motility but also helps to prevent superinfection by strengthening the cortical actin barrier. Our results add to the evidence that the accessory factors of HIV-1 suppress innate immune activation by several mechanisms, including Nef-mediated actin modulation, Vpu-mediated inhibition of NF-κB and tetherin, and inhibition of IRF3/NF-κB by Vpr[68–70].

Our data indicate that actin disturbances arise early during HIV-1 infection, while RLR-mediated viral sensing and its antagonism by Nef primarily occur later in the viral life cycle after proviral integration. Mechanisms suppressing post-integration sensing might be particularly relevant for the ability of HIV-1 to prevent effective antiviral immune responses, since viral nucleic acids are efficiently shielded by the capsid prior to proviral integration[17,71]. In line with this concept, Nef is only found in minimal amounts in HIV-1 virions but is expressed abundantly early after proviral integration and throughout the replication cycle.

We found that Nef prevents recruitment of R12C to both MDA5 and RIG-I in infected primary macrophages (Fig. 5d, Supplementary Fig. 8d). Both RIG-I-like receptors have been reported to sense HIV-1 infection, although accumulating evidence supports a key role of MDA5 in recognizing intron-containing HIV-1 RNA after proviral integration[72–76]. Consistent with this, depletion of MAVS or MDA5 (but not RIG-I) significantly reduced NF-κB activation in THP-1 dual cells (Fig. 4c), and MDA5 knockout was sufficient to suppress *MX1* induction in primary macrophages (Fig. 4g). However, unlike depletion of MAVS, the signaling adaptor for both RIG-I-like receptors, MDA5 depletion alone did not fully abrogate sensing of HIV-1 Nef F191A in THP-1 cells (Fig. 4a, b). Thus, RIG-I may contribute to post-integration sensing of HIV-1 or partially compensate for the lack of MDA5[55,56,77]. RIG-I and MDA5 have overlapping specificity[78] and compensatory effects are well documented[56,77]. Importantly, our data show that Nef targets the actin cytoskeleton and thus R12C, which is required for efficient activation of both RIG-I-like receptors[14]. Thus, by interfering with R12C, Nef suppresses both MDA5- and RIG-I–dependent sensing. However, the relative contribution of each receptor to HIV-1 recognition and its antagonism by Nef requires further investigation.

Previous studies have shown that the interaction of Nef with PAK2 and the resulting inhibition of host cell actin dynamics is conserved between different primate lentiviruses, including SIVcpz, the original host of pandemic HIV-1 group M strains[29,79]. Thus, it is tempting to speculate that suppression of efficient RLR activation is a fundamental property of primate lentiviral Nef proteins. In vivo relevance of Nef-mediated evasion from RIG-I-like receptors' immune recognition is also suggested by evidence that disruption of the Nef-PAK2 association negatively impacts long-term virus spread and pathogenesis of HIV-1 replication in humanized mice[35]. This is even more remarkable considering it may represent a disadvantage in the initial phase of infection[35]. Notably, mutations of the SH3 domain interaction motif (PxxP) to AxxA in the Nef protein of the pathogenic SIVmac239 molecular clone that disrupt its association with PAK2 reverted in the majority of infected rhesus macaques[80,81]. However, in the absence of reversions, animals either rapidly progressed to simian AIDS or controlled the infection. While further studies are required, it is tempting to speculate that an impaired ability of Nef to suppress innate immune activation may either lead to pathogenic excessive inflammation or effective control of lentiviral pathogens. HIV-1 may thus "accept" a transient fitness loss to maintain a potent immune evasion mechanism.

In conclusion, our results demonstrate that the accessory Nef protein stabilizes the interaction of R12C with the actin cytoskeleton to prevent RLR-mediated sensing of post-integration HIV-1 RNAs and induction of effective innate immune defenses. Identification of this immune evasion mechanism may open new avenues for the improvement of AIDS vaccines and cure strategies.

# Methods

## Cell culture
All cells were cultured at 37 °C in a 5% CO$_2$ atmosphere. Human embryonic kidney 293 T cells (HEK293T; ATCC) and HEK293T *PPP1R12C* KO cells[14] were maintained in Dulbecco's Modified Eagle Medium (DMEM) supplemented with 10% heat-inactivated fetal calf serum (FCS), L-glutamine (2 mM), streptomycin (100 µg/mL), and penicillin (100 U/mL) (DMEMxxx). THP-1 (ATCC) and THP-1 Dual cells (Invivogen) were maintained in RPMI supplemented with 10% heat-inactivated fetal calf serum (FCS), L-glutamine (2 mM), streptomycin (100 µg/mL), and penicillin (100 U/mL) (RPMIxxx). To obtain PMA-differentiated THP-1 Dual cells, $0.75 \times 10^6$ cells/mL were seeded in 12-well plates in RPMIxxx 10% with 20 ng/mL PMA. After 3 h, adherent cells were washed once with fresh media and incubated in RPMIxxx 10% for 3 additional days. Chinese hamster ovary cells (CHO) expressing an inducible SF2 *nef* gene, either WT or containing the G2A, AxxA, and F195A mutations, were previously described[28] and cultured in MEMα supplement with 10% heat-inactivated tetracycline-free fetal calf serum (FCS), L-glutamine (2 mM), streptomycin (100 µg/mL), and penicillin (100 U/mL). Jurkat cells with the large T antigen of simian virus 40 (JTAg) were cultured with RPMI medium 1640 (1x) + GlutaMAXTM-l (Gibco) supplemented with 10% heat-inactivated fetal bovine serum advanced (Capricorn, FBS-HI-11A) and 1% penicillin–streptomycin (Gibco).

## Primary cell cultures
PBMCs were isolated from buffy coats from healthy donors using the Pancoll separating solution. Prior to infection, PBMCs were activated with 1 µg/mL PHA. CD4+T cells were obtained by negative selection by mixing RosetteSep Human CD4+ T Cells Enrichment Cocktail (Stem Cell, Cat#15022) with the buffy coat prior to density centrifugation. Residual red blood cells were lysed for 5 min at room temperature using ACK lysis buffer (Lonza #BP10-548E). CD4+ T cells were kept in RPMIxxx supplemented with 10 ng/mL IL-2 (Miltenyi Biotec #130-097-745) and activated for three days with αCD3/αCD28 beads prior to infection. For MDMs, PBMCs were cultured in RPMIxxx supplemented with 10 ng/mL M-CSF (R&D Systems, Cat#416-ML) and seeded at a density of $3 \times 10^6$ cells/well in 12-well NUNC plates. Cells were differentiated for three days before media change to RPMIxxx without M-CSF and cultured for another 4 days prior to infection. Monocytes were isolated from PBMCs with negative selection using EasySep Human Monocyte Enrichment Kit without CD16 Depletion (Stem Cell

Cat#100-1525) according to the manufacturer's instructions. Monocyte-derived dendritic cells (moDCs) were differentiated from isolated monocytes and seeded in 12-well Nucleon Delta plates in IMDMxxx 10% FCS, sodium pyruvate, 800 U/mL GM-CSF (R&D Systems, Cat#215-GM), and 250 U/mlL IL-4 (R&D Systems, Cat#BT-004) for 2 days. Non-adherent cells were washed out by media change and further differentiated for 4 days without cytokines.

## Expression constructs

CH042 (TF), NL4.3 (X4), and NL4.3 *nef* stop proviral constructs were previously described. R191A or F191A mutations in the *nef* ORF were inserted in HIV-1 proviral constructs, CH042 and NL4-3, by overlap PCR using outside primers binding in BamHI and XbaI or PacI and MluI restriction sites before the *nef* ORF and in the 3′ LTR (all oligonucleotide sequences are listed in the Supplementary Data 1). PCR products were cloned into a linearized vector using the InFusion HD cloning kit (Takara, Cat#102518). G2A and AxxA mutations were inserted by using the Q5 Site-Directed Mutagenesis Kit (NEB, Cat #E0554). All mutations were confirmed by Sanger Sequencing. VSVg-expressing plasmid was previously described[82].

## CRISPR-Cas9 KO in primary cells

CRISPR-Cas9 KO in primary cells was performed as previously described[83]. In brief, for CD4+T cells, $1 \times 10^6$ primary CD4+T cells (stimulated) were transfected with the HiFi Cas9 Nuclease V3 (IDT)/gRNA complex (80 pmol/300 pmol) using an NT control or PAK2 sgRNA using the Amaxa 4D-Nucleofector with P3 Lonza Kit (Lonza Cat#V4XP-3032), pulse code EO115. And cultured in RPMIxxx containing 20% FCS. For MDMs, $1 \times 10^6$ primary monocytes were transfected with the HiFi Cas9 Nuclease V3 (IDT, Cat# 1081060)/gRNA complex (80 pmol/300 pmol) using a non-targeting (NT) control (5′-ACG GAG GCT AAG CGT CGC AA-3′) or MAVS sgRNA (5′-GTA GAT ACA ACT GAC CCT GT-3′) or MDA5 (5′-TCA TGA GCG TTC TCA AAC GA-3′), MX1 (5′-AAT CTT GAC GAA GCC TGA TC-3′), PAK2 (5′-AGT AAT CGA GCC CAC TGT TC-3′) sgRNA using the Amaxa 4D-Nucleofector with P2 Lonza Kit (Lonza, Cat# V4XP-2032), pulse code DK100. Two electroporation reactions were seeded per well of 12-well NUNC plates in IMDM supplemented with 20% heat-inactivated human serum, L-glutamine (2 mM), streptomycin (100 μg/mL), penicillin (100 U/mL), Sodium Pyruvate (100 U/mL) (IMDMxxx), and M-CSF (10 ng/mL) and differentiated for three days. Media was changed for IMDMxxx without M-CSF and culture for another 4 days prior to infection.

## R12C silencing by siRNA transfection

MDMs were transfected after differentiation with 80 nM of siGENOME Human PPP1R12C siRNA SMARTpool (Horizon Discovery, Cat#M-013775-01-0020) or siGENOME Non-Targeting Control siRNA (Horizon Discovery, Cat#D-001210-03-20) using Lipofectamine RNAiMAX transfection reagent (Thermo Fisher, Cat# 13778100) using the manufacturer's protocol. The following day, the medium was changed, and the following day, MDMs were infected as indicated.

## Virus production and HIV stocks

For VSV-G pseudotyped virions, one day prior to transfection, $8.5 \times 10^5$ HEK293T cells/well were seeded in 6 wells. Cells were transfected with calcium chloride with 5 μg of the provirus and 1 μg of the VSVg expression plasmid. The next day, the media was changed for DMEMxxx containing 2.5% of FCS, and 48 h post-transfection, the virus-containing supernatant was harvested, clarified by centrifugation, aliquoted, and frozen at −80 °C. VSV-G-pseudotyped HIV was treated with RQ1 DNAseI (Promega, Cat# M6101) one hour prior to infection at 37 °C. For viruses used in replication and endogenous MDA5 dephosphorylation assays, one day prior to transfection, $10 \times 10^6$ HEK293T cells were seeded in 15 cm dishes. Cells were transfected with 25 μg of the provirus mixed with OptiMEM and 75 μL of LT1

transfection reagent. The next day, the media was changed, and 48 h post-transfection, the virus-containing supernatant was harvested, clarified by centrifugation, and filtered through a 0.45 μm filter to remove cell debris. The virus was pelleted on a 20% sucrose cushion for 2 h at 4 °C in an ultracentrifuge at 96416 g. The pellet was resuspended in DMEMxxx containing 2.5% of FCS, aliquoted, and frozen at −80 °C.

## p24 ELISA

Virus stock or culture supernatants were analyzed for p24 content. Samples were lysed in 10% Triton X-100, and p24 was quantified using HIV-1 Gag p24 DuoSet ELISA (R&D Systems, Cat# DY7360-05) according to the manufacturer's protocol.

## Transduction of macrophages and THP-1 cells

VSV-G pseudotyped viral stocks were normalized to ~5 ng p24/mL in serum-free DMEM (MDMs), serum-free IMDM (MoDCs), or 2% FCS RPMI (THP-1) and added to the cells. For primary cells, cells were incubated overnight at 37 °C and washed the next day with media. Supernatant or cells were harvested 48 h post-transduction. For THP-1 cells, supernatant was harvested daily for analysis until the last time point indicated. For CD4+T cells, cells were spinoculated with 10 ng/mL of virus in 96-well Flat bottom 96 well plate with 500,000 cells/well for 2 h at 37 °C. CD4+T cells were washed three times and incubated in RPMIxxx 10% FCS supplemented with 10 ng/mL IL2.

## Treatment with antiretrovirals and anti-VSV-G antibody

Entry of input particles carrying VSV-G was blocked by adding 10% (v/v) of I1 Hybridoma supernatant (I1, mouse hybridoma supernatant from CRL-2700; ATCC) to the cell culture supernatant and the virus stock for 10 min prior to infection. Antiretroviral drugs (Maraviroc, BEI Resources HIV Reagent Program Cat#HRP-11580, 5 μM; AMD3100, MedChemExpress Cat#HY-10046, 1 μM; Nevirapine, BEI Resources HIV Reagent Program Cat#HRP-4666 1 μM; Raltegravir, BEI Resources HIV Reagent Program Cat#11680 20 μM; and Nelfinavir, BEI Resources HIV Reagent Program Cat# ARP-4621 1 μM) were added to the cells 10 min prior to infection. To block IFN-I signaling, cells were incubated in the presence of 200 ng/mL of B18R (R&D Systems, Cat# 8185-BR) during infection. Antibody, antiretrovirals, and B18R were kept after washing and over the course of the assay.

**Stimulation of HEK293T cells.** HEK293T cells were seeded into 12-well plates. The following day, they were transfected with poly (I:C) HMW/lyovec (1 μg/mL, 200, 40 ng/mL) or G3-YSD/lyovec (1 μg/mL, 200, 40 ng/mL), according to the manufacturer's protocol (Invivogen). For Sendai virus infection, cells were infected with 25, 5, and 1 HAU/mL. For cGAMP stimulation, cells were treated with 20 μg, 4, and 0.8 μg/mL. 24 h after treatment, cells were harvested for RNA isolation.

## Nucleoside-assisted infections

Stock solutions of dNTP precursors 2′-deoxyadenosine, 2′-deoxyguanosine, thymidine (2′-deoxyadenosine; 2′-deoxyguanosine, thymidine and 2′-deoxycytidine hydrochloride, Sigma Aldrich, Cat# D7400, 854999, T1895, D0776) were prepared in RPMI-HCl (pH 4.1) at 40 mM for 2′-deoxyadenosine and thymidine and 20 mM for 2′-deoxyguanosine. 2′-deoxycytidine hydrochloride stock solution was prepared in RPMI at 20 mM. 0.5 or 2 mM of each dNTP precursor was preheated at 37 °C with shaking to dissolve and was added to the culture medium 10 min prior to infection, during infection, and after media change as previously reported[84].

## qRT-PCR

*MX1*, *CXCL10*, *CCL5*, and *IFNB1* mRNA levels were determined in infected cells collected at 48 h post-infection or HEK293T treated as indicated for 24 h. For MDMs infected with non-pseudotyped CH042, cells were harvested at 96 h post-infection. Total RNA was isolated using the RNeasy PlusRNA Mini Kit (Qiagen, Cat#74136) according to

the manufacturer's instructions. qRT-PCR was performed according to the manufacturer's instructions using TaqMan Fast Virus 1-Step Master Mix (Thermo Fisher, Cat# 5555532) and a OneStepPlus Real-Time PCR System (96-well format, fast mode). Predesigned primer and probe mixes were purchased from Thermo Fisher and multiplexed with GAPDH (Thermo Fisher Scientific (4310859) as an internal control for normalization. To measure induction of *mx1-like* in Nef inducible CHO cells, Chinese hamster-specific primers and probes were used (mx1-like, Thermo Fisher Scientific Cat#Cg04463034_g1 and gapdh Cat# 4308313). All reactions were run in duplicates.

### Integration Alu PCR

Cells were lysed with RLT plus buffer containing 1% β-mercaptoethanol, and RNA and DNA were isolated with the AllPrep DNA/RNA isolation kit (Qiagen, Cat# 80204). In a first step, pre-amplification, Alu-PCR was performed on 20 ng of extracted DNA. Reaction set up was as follows: 0.25 μL of each primer (Alu F, 5′-GCC TCC CAA AGT GCT GGG ATT ACA G-3′, Gag R, 5′-GCT CTC GCA CCC ATC TCT CTC C-3′, 10 μM), 1 μL dNTP, 5 μL Buffer, 10 μL Q-Buffer, 0.5 μL Taq DNA Polymerase (Qiagen), and 13 μL of water. Cycling conditions were set as 95 °C 5 min (95 °C 30 s, 55 °C 15 s, 68 °C 4 min) x 25 cycles, 68 °C 10 min. qRT-PCR was performed with 2.5 μL of the PCR reaction product with PCR with TaqMan Fast Univ. PCR Master Mix (Applied Biosystems) and the primer/probe set: LTR-R Forward 5′-GCC TCA ATA AAG CTT GCC TTG A-3′, LTR-U5 Reverse 5′-TCC ACA CTA CCA AAA GGG TCT GA-3′, 5′(FAM)-CCC GTC TGT TGT GTG ACT CTG GTA ACT AG-(TAMRA)-3′.

### Flow cytometry

To monitor infection levels of THP-1 Dual cells and MDMs, cells were harvested in 96-well plates, washed once with PBS, and stained for 30 min at RT in the dark with anti-CD4 antibody (PerCP-Cy5.5, Biolegend Cat #317428, 1:50 in PBS) and eBioscience Fixable viability dye 780 (ThermoFisher Scientific Cat#65-0865-14, 1:1000 in PBS). Afterwards, cells were washed twice with PBS and permeabilized 20 min with BD Cytofix/Cytoperm Fixation/Permeabilization Solution Kit (BD Biosciences, Cat#554714) at RT. Cells were washed twice with 1X Perm/Wash solution and stained 30 min at RT with anti-HIV-1 p24 (RD1/PE, Beckman Coulter Cat#6604667, 1:100 in 1X Perm/Wash solution). In cases where MX1 was stained in infected MDMs, anti-Mx1 (Alexa Fluor 647, Abcam Cat# ab237299, 1:500 in PBS) was added during intracellular staining. After washing twice with 1X Perm/Wash solution, cells were fixed in 2% PFA for 30 min at 4 °C. To determine levels of phosphorylated cofilin (at S3) in infected cells, cells were harvested and permeabilized, and fixed immediately in Cytofix/Cytoperm, washed as described, and stained with anti-p24 (RD1/PE, Beckman Coulter Cat#6604667, 1:100 in 1X Perm/Wash solution) and rabbit anti-pS3-Cofilin (Cell Signaling Technology, Cat#3313). After incubation for 30 min at RT, cells were washed and stained with secondary goat anti-rabbit (Alexa Fluor 647, Thermofisher). Cells were acquired with a BD FACSCanto II Flow Cytometer (BD Biosciences).

### Legenplex ELISA

The Legendplex ELISA (Human Anti-Virus Response Panel V02, Biologend Cat#741270) was performed according to the manufacturer's instructions. In brief, supernatants of MDMs or moDCs infected with HIV-1 CH042 WT or expressing R191A were collected 48 h post-infection and incubated for 2 h at room temperature with antibody-coated beads, followed by washing and incubation with the detection antibodies. After incubation with the staining reagent, the beads were fixed in PFA 2% and analyzed in a high-throughput sampler via flow cytometry (Cytoflex, Beckman). Absolute quantification was performed using a standard and the Biolegend Legendplex v8.0 software.

### Whole-cell lysates

Whole-cell lysates were prepared by harvesting cells in phosphate-buffered saline (PBS, Gibco, Cat#14190144). If not mentioned otherwise, the cell pellet (500×*g*, 4 °C, 5 min) was lysed in transmembrane lysis buffer (150 mM NaCl, 50 mM 4-(2-hydroxyethyl)-1-piperazineethanesulfonic acid (HEPES) pH 7.4, 1% Triton X-100, 5 mM ethylenediaminetetraacetic acid (EDTA)) by vortexing at maximum speed for 30 s. For assessing phosphorylation levels of RLRs, cells were treated for 25 min with 50 nM calyculin A prior to harvest and cells were lysed in NP40 buffer (50 mM HEPES pH 7.4, 150 mM NaCl, 1% (v/v) NP-40, 1 mM EDTA) and in the presence of protease inhibitor cocktail, phosphatase inhibitor cocktail 3 and calyculin A. Cell debris was removed by centrifugation (20,000×*g*, 4 °C, 20 min), and the protein concentration of the supernatants was quantified using a BCA assay (Pierce Rapid Gold BCA Protein Assay Kit, Thermo Fisher Scientific, Cat#A53225). The lysates were then used immediately or stored until analysis at −20 °C.

### SDS−PAGE and immunoblotting

SDS−PAGE and immunoblotting were performed using standard techniques. In brief, whole-cell lysates were mixed with 4x protein sample loading buffer (at a final dilution of 1x) supplemented with 10% β-mercaptoethanol, heated to 95 °C for 5 min, separated on NuPAGE 4−12% Bis−Tris Gels (Invitrogen, Cat#NP0321BOX) for 90 min at 90 V and blotted onto Immobilon-FL PVDF membranes (Merck Millipore, Cat#05317-10EA). For analysis of ISGylation of MDA5, samples were run on a 7% separating gel. The transfer was performed with a semi-dry transfer apparatus at a constant voltage of 30 V for 30 min. After the transfer, the membrane was blocked in 1% Casein in PBS. Proteins were stained with primary antibodies mouse anti-ISG5 (1:500, Santa Cruz Biotechnology, Cat#sc-166755), rabbit anti-PAK2 (1:1000 Cell Signaling Technology, Cat#2608), rabbit anti-MAVS (1:1000, Cell Signaling Technology), rabbit anti-MDA5 (1:1000, Cell Signaling Technology, Cat#5321), rabbit anti-pS88 MDA5[14], mouse anti-RIG-I (1:1000, Adipogen, AG-20B-0009-C100), Rabbit anti-AU1, (1:1000, Novus, Cat# NB600-453), rat anti-GAPDH (1:1000, BioLegend, Cat#607902), and subsequently Infrared Dye labeled secondary antibodies (LI-COR, IRDye 680RD Goat anti-Mouse IgG Secondary Antibody, 926-68070; IRDye 680RD Goat anti-Rabbit IgG Secondary Antibody, Cat#926-68071; IRDye 800CW Goat anti-Mouse IgG Secondary Antibody, Cat#926-32210; IRDye 800CW Goat anti-Rabbit IgG Secondary Antibody, Cat#926-32211; IRDye 800CW Goat anti-Rat IgG Secondary Antibody, Cat#926-32219), diluted in 0.05% Casein in PBS. Band intensities were quantified using Image Studio Lite (LI-COR).

### CHO cells expressing Nef

CHO cells were treated for 2 days with 1000, 250, or 50 ng/mL of doxycycline to achieve a similar level of Nef-GFP expression and infected with 25 hemagglutination units (HAU)/mL of Sendai Virus (SeV) for 24 h before the cells were harvested.

### Fluorescence microscopy

Primary MDMs were seeded and differentiated on coverslips in 24-well plates and treated as indicated. The samples were washed with PBS and fixed in 4% paraformaldehyde solution (PFA) for 20 min at RT, permeabilized and blocked with PBS containing 0.05% Triton X-100 and 5% FCS for 1 h at RT. Afterwards, the cells were washed with PBS and incubated for 2 h at 37 °C with mouse anti-R12C (1:1000, Santa Cruz Biotechnology Cat#sc-398415) in PBS/1% FCS. After washing with PBS/0.1% Tween 20, the samples were incubated with the secondary antibody donkey anti-mouse IgG (H + L) Alexa Fluor Plus 568 (1:500, Thermo Fisher Cat# A10037), phalloidin-647 (1:500, Thermo Fisher Cat# A22287), mouse anti-p24 (FITC, 1:100, Beckman Coulter Cat#6604665) and 500 ng/mL DAPI (Sigma-Aldrich) for 2 h at 4 °C in the dark. Next, the samples were washed with PBS/0.1% Tween 20 and

water, and the coverslips were mounted onto microscopy slides. Images were acquired using a Zeiss LSM 710 confocal laser scanning microscope with ZEN 2010 imaging software (Zeiss). Images were analyzed with ImageJ (Fiji). Co-localization was determined by calculating the Pearson coefficient.

## Analysis of cell morphology and actin polymerization in response to T cell activation

$5 \times 10^6$ JTAgs were co-transfected with 3 µg of the pEGFP plasmid and with 20 µg of the pTWIST vector control or a Nef expression plasmid by electroporation (250 V, 850 µF) using the GenePulser Xcell from Bio-Rad. The ability of Nef to interfere with actin dynamics and morphological changes induced by T cell activation was assessed as described previously[34]. Briefly, coverslips were treated with 1:10 in water solution of 0.1% poly-L-lysine (Sigma Aldrich, Cat#P4707) for 10 min at room temperature, coated with 10 µg/mL purified NA/LE mouse anti-human CD3 (BD Biosciences, Cat#555336) and 10 µg/mL purified NA/LE mouse anti-human CD28 (BD Biosciences, Cat#567117) antibody solution in PBS overnight at 4 °C, rinsed with PBS and stored at 4 °C until further use. $3 \times 10^5$ co-transfected JTAgs were seeded onto the stimulatory coverslips and incubated for 5 min at 37 °C. The cells were fixed with 3% paraformaldehyde for 15 min, permeabilized with 0.1% Triton-X100 (Sigma Aldrich, Cat#9036-19-5) in PBS for 5 min, and stained with 1:1000 rhodamine phalloidin (Invitrogen, Cat#R415) solution in 3% BSA (Roth, Cat#8076.2) in PBS for 1 h to visualize F-actin. Coverslips were mounted on glass slides using Fluoromount-G™ with DAPI (Invitrogen, Cat#00-4959-52). The cell spreading and F-actin ring formation were assessed using epifluorescence microscopy (Olympus IX81 S1F-3, cellM software) with a UPlanApo 60x/1.40 oil immersion objective. For each of the independent triplicate experiments, 100 GFP+ cells per sample were analyzed, and representative images were taken using a CREST spinning disk with a Nikon Ti2 microscope with a PlanApo λD ×100/1.45 OFN25 DIC N2 oil immersion objective and 477 nm as well as 546 nm excitation lasers.

## Immunoprecipitation

Primary MDMs were either infected with VSV-G-pseudotyped HIV-1 CH042 WT, HIV-1 CH042 expressing Nef R191A, or treated with poly (I:C) HMW for 48 h in the presence of 0.5 mM nucleosides. Subsequently, WCLs were prepared, and input samples were saved for Western blotting. The WCLs were then incubated with anti-MDA5-conjugated dynabeads (Thermo Fisher, Cat#10003D) at 4 °C for 4 h under continuous rotation. Afterward, the beads were washed three times with transmembrane lysis buffer before being incubated in 1x Protein Sample Loading Buffer supplemented with 15% β-mercaptoethanol. Samples were then heated at 95 °C for 10 min and subjected to SDS–PAGE followed by immunoblotting. THP1 cells (from ATCC) were infected with HIV-1 NL4-3 viruses (WT, F191A, or nef mutant). At 20 h post-infection, cells were treated with CalA (Invitrogen, PHZ1014) for 25 minutes and then harvested for preparation of whole cell lysates. Immunoprecipitation of MDA5 from WCL was performed by using anti-MDA5 antibody (Proteintech #21775-1-AP) followed by Protein G Dynabeads (ThermoFischer #10004D).

To control for KO in HEK293T *R12C* KO cells and THP-1 dual *CGAS* KO cells, cells were lysed as previously and incubated overnight with 50 µL of dynabeads protein G (Thermo Fisher, Cat#10003D) pre-conjugated with 10 µL of either the anti-R12C rabbit antibody or the anti-cGAS antibody (Proteintech, Cat#26416-1-AP). Afterward, the beads were washed three times with transmembrane lysis buffer before being incubated in 1× Protein Sample Loading Buffer supplemented with 15% β-mercaptoethanol. Samples were then heated at 95 °C for 10 min and subjected to SDS–PAGE followed by immunoblotting.

## MDA5 ISGylation in HEK293T cells

MDA5 ISGylation assay was performed as previously described[51]. Briefly, HEK293T cells were transfected for 24 h with 1 µg FLAG-MDA5, 0.25 µg V5-ISG15, 1 µg HA-Ube1L (E1), 0.25 µg FLAG-UbcH8 (E2), and 1 µg Myc-HERC5 (E3) together with either empty vector (−) or AU1-tagged Nef WT or F191A (0.5 or 1 µg). Cells were then stimulated with 1 µg/mL poly(I:C)-HMW for 12 h before harvesting. ISGylation of FLAG-MDA5 was assessed by immunoprecipitation with FLAG-M2 beads and IB with anti-V5 and anti-FLAG.

## Proximity ligation assay

Proximity ligation assay (PLA) was performed as previously described[85]. In brief, primary MDMs were seeded and differentiated on coverslips in 24-well plates and treated as indicated. At 48 hpi, cells were washed once with cold PBS and fixed with 4% PFA. For staining following antibodies were used: anti-PPP1R12C (rabbit, previously described in ref. [14]), anti-RIG-I (Adipogen, Cat# AG-20B-0009-C100), anti-PPP1R12C (Santa Cruz Biotechnology, Cat# sc-398415), and anti-MDA5 (Proteintech, Cat# 21775-1-AP). Images were acquired on a Zeiss LSM 710 and processed using ImageJ (Fiji).

## Luciferase and SEAP reporter assays

Supernatants of infected THP-1-Dual cell lines were harvested at the indicated time points, and luciferase activities of the lucia-luciferase or SEAP activity were determined. ISRE-lucia luciferase activity was measured 1 s after injecting 20 mM coelenterazine (PFK Biotech, Cat#102173) and NF-κB-SEAP activity via Quanti Blue Substrate (Invivogen, Cat#rep-qb) and fixed in 1% PFA. Luciferase measurements were performed using an Orion II microplate Luminometer and the Simplicity software (Berthold), and SEAP activity was measured at 650 nm by using a Vmax kinetic microplate reader (Molecular Devices) and the SoftMax Pro 7.0.3 software.

## Quantification and statistical analysis

Statistical analyses were performed using GraphPad Prism 9. *P*-values were determined using a two-tailed Student's *t*-test with Welch's correction, ratio paired *t*-test, or repeated measure two-way ANOVA with Geisser−Greenhouse's correction, as indicated. Unless otherwise stated, data are shown as the mean of at least three biological replicates ± SEM. Significant differences are indicated as: *$p < 0.05$; **$p < 0.01$; ***$p < 0.001$; ****$p < 0.0001$. Not significant differences are usually not indicated. Specific statistical parameters are specified in the figure legends.

## Ethical statement for human samples

Experiments involving human cells isolated from human blood were reviewed and approved by the Institutional Review Board (i.e., the Ethics Committee of Ulm University). Individuals and/or their legal guardians provided written informed consent prior to donating blood. All human-derived samples were anonymized before use.

## Reporting summary

Further information on research design is available in the Nature Portfolio Reporting Summary linked to this article.

## Data availability

All data generated in this study are provided in the Source Data file. Source data are provided with this paper.

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

## Acknowledgements

We would like to thank Martha Meyer, Daniela Krnavek, Jana-Romana Fischer, Alisia Elsesser, Regina Burger, Kerstin Regensburger, and Dré van der Merwe for excellent technical assistance. F.K. was supported by grants from the German Research Foundation (DFG CRC 1279 and CRC 1506), as well as an ERC Advanced Grant (TraitorViruses). K.M.J.S. acknowledges funding by the German Federal Ministry of Education and Research (BMBF; IMMUNOMOD-01KI2014), the DFG (CRC 1279, SP 1600/7-1, SP 1600/13-1). M.H. is supported by a Bausteine grant of Ulm University. O.T.F. acknowledges funding by the DFG (project 538258788, FA 378/27-1).

## Author contributions

A.L. and C.P.B. performed the majority of the experiments and analyzed the datasets. D.A., A.d.L., M.H., J.Z., C.M.R.-Q., and M.V. performed additional experiments. B.S., O.T.F., and M.U.G. provided scientific suggestions and resources, and edited the manuscript. All authors reviewed the manuscript. K.M.J.S. and F.K. conceived and designed the project and wrote the manuscript.

## Funding

## Competing interests

The authors declare no competing interests.
