## [Transparent Peer Review file · Nature Communications]

Nef stabilizes actin to prevent HIV-1 sensing by RIG-I-like receptors

Corresponding Author: Professor Frank Kirchhoff

Version 0:

Reviewer comments:

Reviewer #1

(Remarks to the Author)

In this revised manuscript, the authors go some way towards addressing prior reviewer comments on the original Nature Microbiology submission. However, in my view, the authors choose to avoid addressing several important points and the overall novelty of the findings do not seem adequate for a journal of the quality of Nature Communications.

Main Points:

The authors choose to talk around rather than directly addressing several prior concerns. For example, comments on the quality of IP data in figure 4e were completely ignored, validation of the antibodies was only partial and mostly focused on potential cross-reaction with p24 rather than specificity of the PLA signal, and comments on the quality of actin analysis was addressed by some indirect measure of Nef effects on T-cell activation, which is not relevant to the question. The authors try to explain why they jump between many different cell lines but this is still problematic and not an ideal approach, sometimes feeling like choosing a cell line that produces the desired effect more than anything.

Reviewer 2 raises a valid point that the authors rely on a single clone of an R12C knockout line. The authors just respond that overexpression is problematic but made no attempt to either reduce expression levels or use an alternative approach. It would seem quite easy to use RNAi depletion, with multiple independent siRNAs for example. Reviewer 2's comments on the use of VSV-G are also quite valid.

In Fig R1 in the response letter, the authors argue that variability in the WT samples between panels 1a and 1b are insignificant, but the fold mRNA induction of the WT and R19A samples from Fig 1a and 1b seems reversed and different to the original version – both the magnitudes and the number and distribution of data points seem very different for some samples. For example, mean WT in 1a was over 2, it's now 1 in figure R1, and for R191A in figure R1 there are two data points below 2 for 1a but in the original 1a there were none. These do not seem to be a direct comparison of the data sets that were questioned. In general, the authors tend to obfuscate points and responses, which makes reviewing their papers quite exhausting when reviewers are simply trying to work with authors to improve their manuscript.

In response to comments such as the effects are small, the authors simply argue they are robust. This seems subjective but overall, the findings largely link previous discoveries and do not seem to be sufficiently novel or robust for a journal such as Nature Communications.

Reviewer #2

(Remarks to the Author)

In this revised version of the manuscript, the authors present an impressive amount of new data that address my previous concerns to a considerable extent. Importantly, in response to the criticism that the effect of knocking out MDA5 in THP-1 cells on the sensing of a Nef mutant was rather modest, they now show in Fig. 5f and g that knocking out MDA5 has a more significant effect in primary monocyte-derived macrophages. However, a caveat here is that the KO cells appear to be

misabeled in Fig. 5g (MAVS KO instead of MDA5 KO). The manuscript text (lines 238-239) suggests that this is a labeling error, but this needs to be clarified.

Additional comments:

1. The authors attempted to address the concern that VSVG pseudotyping may have affected the results by showing data obtained with non-pseudotyped HIV-1. In particular, Fig. 1i now indicates that HIV-1 with a mutant Nef (R191A) unable to engage PAK2 induced CXCL10 to a larger extent than the WT virus. However, under the same conditions, the effects on cellular MX1 mRNA levels shown in supplementary Fig. 2i were rather modest, as the authors acknowledge. It remains unclear whether the effect of the R191A mutant was significant, because a statistical analysis is lacking. This should be provided.
2. Fig. 4a,b: It is evident in this figure that knocking out MAVS had a profound effect on the induction of the ISRE-luc reporter, whereas the effect of knocking out MDA5 was quite modest. The authors now suggest in the revised version that RIG-I may partially compensate for the lack of functional MDA5. Perhaps this could be tested by knocking out RIG-I in the MDA5 KO cells used in Fig. 4b.
3. Attempts to perform rescue experiments with R12C to ensure that the results in Fig. 5b were not due to clonal variation or an off-target effect were unsuccessful. Therefore, the conclusion that R12C is required for the induction of IFNB1 mRNA still rests on experiments performed with a single clone of R12C KO cells. In my opinion, the authors should repeat these experiments with several additional KO clones, or with a polyclonal population of KO cells.
4. Line 280: "Fig. 5e" should be Fig. 5d.

Version 1:

Reviewer comments:

Reviewer #2

(Remarks to the Author)

In this revised submission to Nature Communications, the authors have performed additional experiments that strengthen their conclusion that R12 is crucial for the sensing of HIV-1 in their system. They also provide additional evidence that authentic (non-pseudotyped) HIV-1 with a Nef that is defective for PAK2 binding can induce the expression of host proteins involved in innate immunity.

However, because double-KO cells were not viable when infected with HIV-1, the authors were unable to examine whether compensation by RIG-I accounts for the relatively modest effect of knocking out MDA5 in Fig. 4b, which contrasts sharply with the profound effect of knocking out MAVS in Fig. 4a. Have the authors considered to use RNAi to knock down RIG-I in the MDA5 KO cells as an alternative approach to address this important issue?

Also, the authors' model in Supplementary Fig. 9 appears to suggest that the actin disturbance that triggers the release of R12C and viral sensing occurs as a consequence of the penetration of the viral core through the cortical actin cytoskeleton. It seems there is something missing from this model, because the R12C-dependent sensing was dependent on proviral integration.

Reply to the reviewer`s comments (in italic letters)

Reviewer #1 (Remarks to the Author): In this revised manuscript, the authors go some way towards addressing prior reviewer comments on the original Nature Microbiology submission. However, in my view, the authors choose to avoid addressing several important points and the overall novelty of the findings do not seem adequate for a journal of the quality of Nature Communications.

We previously addressed novelty, corrected his/her factual inaccuracies, and added a total of 28 new figure panels to address technical concerns. As specified below, reviewer 1 apparently overlooked much of our previous response and missed that some experiments directly addressed specific points raised.

Main Points:

The authors choose to talk around rather than directly addressing several prior concerns. For example, comments on the quality of IP data in figure 4e were completely ignored, validation of the antibodies was only partial and mostly focused on potential cross-reaction with p24 rather than specificity of the PLA signal, and comments on the quality of actin analysis was addressed by some indirect measure of Nef effects on T-cell activation, which is not relevant to the question. The authors try to explain why they jump between many different cell lines but this is still problematic and not an ideal approach, sometimes feeling like choosing a cell line that produces the desired effect more than anything.

The IP shows conjugation of endogenous ISG15 to endogenous MDA5 in HIV-1 infected primary macrophages. Thus, signals are weaker and less clean compared to commonly used overexpression assays but also of higher physiological relevance. To further address this, we now show that the ISG15 signal on immunoprecipitated MDA5 is significantly increased in primary macrophages infected with the R191A Nef mutant virus with quantification of blots from additional donors (new Fig. 4i, right). We also confirmed the results by transient transfection of HEK293T cells (new Fig. 4h).

Lack of cross-reactivity was excluded to specifically address the previous comment of this reviewer "Have the authors confirmed that the R12C antibody does not cross-react with p24?". Our previous data showed that the antibodies used are highly specific. KD of R12C in primary macrophages now further confirms specificity of the previously characterized R12C antibody (Acharya et al, Cell, 2022) used for the confocal microscopy images (new suppl. Fig. 8b). To further address specificity, we show that PLA signals are absent if only one of the two antibodies is used for PLA (new supplement Fig. S8a and updated suppl. Fig. 8d). Finally, we now show that PLA signal is drastically reduced after R12C KD in primary macrophages (new suppl. Fig. 8b-c).

We suspect that "indirect measure of Nef effects on T-cell activation" refers to our data showing the effects of Nef on the morphology and actin cytoskeleton organization of CD4+ T cells. As previously mentioned, this experimental model is well established (Stolp et al., 2009, CHM, Stolp et al 2010 JVI; Imle et al. 2015 mBio) and demonstrated that the association of Nef with PAK2 results in the hyperphosphorylation of cofilin and impairs actin polymerization. Our results show that these activities are similarly impaired by the R191A and F191A mutations in the CHO42 and NL4.3 Nef variants, respectively. This directly addresses the effects of Nef on actin remodeling in a cell system relevant to the current study.

As previously mentioned, most assays were performed in HIV-1-infected primary macrophages, and we provided clear rationales for the different experimental systems. Since Reviewer 1 did still not specify which approaches were considered "not ideal," no further changes were made.

Reviewer 2 raises a valid point that the authors rely on a single clone of an R12C knockout line. The authors just respond that overexpression is problematic but made no attempt to either reduce expression levels or use an alternative approach. It would seem quite easy to use RNAi depletion, with multiple independent siRNAs for example. Reviewer 2's comments on the use of VSV-G are also quite valid.

Reviewer 2 noted that “we present an impressive amount of new data that address my previous concerns to a considerable extent”. The VSV-G issue had already been addressed in our earlier response; to resolve the remaining point, we confirmed our results in a second R12C knockout cell line (new suppl. Fig. 7e, f).

In Fig R1 in the response letter, the authors argue that variability in the WT samples between panels 1a and 1b are insignificant, but the fold mRNA induction of the WT and R19A samples from Fig 1a and 1b seems reversed and different to the original version – both the magnitudes and the number and distribution of data points seem very different for some samples. For example, mean WT in 1a was over 2, it's now 1 in figure R1, and for R191A in figure R1 there are two data points below 2 for 1a but in the original 1a there were none. These do not seem to be a direct comparison of the data sets that were questioned. In general, the authors tend to obfuscate points and responses, which makes reviewing their papers quite exhausting when reviewers are simply trying to work with authors to improve their manuscript.

We made a copy-paste error in the figure provided to this reviewer and apologize for it. Please see the correct version below. Indeed, MX1 mRNA levels increased moderately in WT infected primary macrophages in Fig. 1a but not in Fig. 1b. However, some variation in primary cells is expected. More importantly, HIV-1 encoding the R191A Nef generally induced significantly higher levels compared to WT HIV-1. Combined data show that the difference between R191A and WT Nef is highly significant if data are combined, even without pairing the donors for statistical analysis ($p=0.005$). When the donors are paired, the statistical significance is further increased ($p<0.0001$).

Fig.R1 Comparison of immune activation in data obtained in different donors from Fig. 1a and Fig. 1b. a, comparing MDMs infected with CH042 WT. **b**, comparing MDMs infected with CH042 expressing R191A Nef, **c**, combining donors from panels a and b, **d**, combining donors from panels a and b and linking donors. Statistical analysis was done using Welch t-test. * $p < 0.05$; *** $p < 0.001$.

We acknowledge the reviewer's “exhausting” efforts. However, to improve manuscripts specific suggestions are more helpful than vague opinions.

In response to comments such as the effects are small, the authors simply argue they are robust. This seems subjective but overall, the findings largely link previous discoveries and do not seem to be sufficiently novel or robust for a journal such as Nature Communications.

The reviewer refers to the first five words of our comprehensive (192 words) response. In brief, we mentioned that effects are also statistically significant. Since this reviewer stated that effects were usually 2-fold, we clarified that magnitudes were ranging from 2.5- to 15-fold and indicated this in the revised figures. We also explained that differences from previous studies stem from methodological variations, and provided a new Supplementary Fig. 3 showing that under artificially enhanced infection conditions, innate sensing increases but key differences between genuine viral strains in primary human cells are obscured. Thus, this point has been addressed.

Reviewer #2 (Remarks to the Author): In this revised version of the manuscript, the authors present an impressive amount of new data that address my previous concerns to a considerable extent. Importantly, in response to the criticism that the effect of knocking out MDA5 in THP-1 cells on the sensing of a Nef mutant was rather modest, they now show in Fig. 5f and g that knocking out MDA5 has a more significant effect in primary monocyte-derived macrophages. However, a caveat here is that the KO cells appear to be mislabeled in Fig. 5g (MAVS KO instead of MDA5 KO). The manuscript text (lines 238-239) suggests that this is a labeling error, but this needs to be clarified.

We thank reviewer 2 for acknowledging the “impressive amount of new data” that address most previous concerns and the constructive suggestions. We corrected the labeling error in Fig. 4g (Fig. 5 was correct and doesn’t have a panel g).

Additional comments:

1. The authors attempted to address the concern that VSVG pseudotyping may have affected the results by showing data obtained with non-pseudotyped HIV-1. In particular, Fig. 1i now indicates that HIV-1 with a mutant Nef (R191A) unable to engage PAK2 induced CXCL10 to a larger extent than the WT virus. However, under the same conditions, the effects on cellular MX1 mRNA levels shown in supplementary Fig. 2i were rather modest, as the authors acknowledge. It remains unclear whether the effect of the R191A mutant was significant, because a statistical analysis is lacking. This should be provided.

Infection rates of primary macrophages by genuine HIV-1 strains are very low; thus, bulk analyses underestimate the magnitude of the effects. As indicated in the revised paper, the difference in MX1 levels was thus not significant. To strengthen our findings, we confirmed that non-pseudotyped R191A-Nef HIV-1 infection also induced higher levels of CXCL10 in primary macrophages isolated by a different protocol and observed a trend for RSAD2 (Viperin).

Fig. R2. Immune induction in MDMs by genuine HIV-1 CH042 infection. CXCL10 and RSAD2 induction measured by qRT-PCR from MDM differentiated from purified monocytes (With STEMCELL EasySep™ Human Monocyte Enrichment Kit without CD16 Depletion) and infected with WT or R191A-Nef CH042 at 4 dpi. Each point represents an individual donor ± SEM (N=5 donors). Statistical analysis was done with ratio paired t-test (*, p<0.05).

2. Fig. 4a,b: It is evident in this figure that knocking out MAVS had a profound effect on the induction of the ISRE-luc reporter, whereas the effect of knocking out MDA5 was quite modest. The authors now suggest in the revised version that RIG-I may partially compensate for the lack of functional MDA5. Perhaps this could be tested by knocking out RIG-I in the MDA5 KO cells used in Fig. 4b.

We performed this experiment. However, the combination of KO followed by HIV-1 infection compromised viability of THP-1 dual cells precluding meaningful analysis.

3. Attempts to perform rescue experiments with R12C to ensure that the results in Fig. 5b were not due to clonal variation or an off-target effect were unsuccessful. Therefore, the conclusion that R12C is required for the induction of IFNB1 mRNA still rests on experiments performed with a single clone of R12C KO cells. In my opinion, the authors should repeat these experiments with several additional KO clones, or with a polyclonal population of KO cells.

To address this, we confirmed our findings in an independently generated R12C KO cell clone (new suppl. Fig. 7d-f). Furthermore, we performed KD of R12C in HEK293T before their transfection. While siRNA transfection generally reduced the induction of IFNB1 after proviral transfection, the induction by NL4.3 Nef F191A transfection was abolished upon KD of R12C.

Fig. R2. R12C a, qRT-PCR analysis showing mRNA levels of PPP1R12C in HEK293T 2 days after siRNA transfection. Statistical analysis was done using Welch's t-test. **b**, qRT-PCR analysis showing mRNA levels of IFNB1 in HEK293T from a, two days after provirus transfection. (N=3 independent experiments). Statistical analysis was done using ratio paired t-test.

4. Line 280: "Fig. 5e" should be Fig. 5d.

Corrected

Reply to the reviewer`s comments (in italic letters)

We thank reviewer 2 for the constructive comments and addressed the two remaining points by textual changes.

1. *However, because double-KO cells were not viable when infected with HIV-1, the authors were unable to examine whether compensation by RIG-I accounts for the relatively modest effect of knocking out MDA5 in Fig. 4b, which contrasts sharply with the profound effect of knocking out MAVS in Fig. 4a. Have the authors considered to use RNAi to knock down RIG-I in the MDA5 KO cells as an alternative approach to address this important issue?*

We considered it. However, RNAi itself can trigger innate immune activation and is often not very efficient. We expanded the discussion section (lines 343-357) to address this point. In brief, we clarified that while both RIG-I and MDA5 have been implicated in HIV-1 sensing, accumulating evidence and our own data indicate that MDA5 play a major role in post-integration sensing. The ability of RIG-I and MDA5 to compensate for one another is well documented and consistent with our finding that KO of MAVS (which disrupts signaling from both receptors) has more severe effects than KO of MDA5 alone. Importantly, we show that Nef prevents recruitment of R12C to both MDA5 and RIG-I in infected primary macrophages and thus inhibits sensing by both receptors. However, as specified in the revised discussion, their relative contribution to HIV-1 recognition and its antagonism by Nef needs further study.

2. *Also, the authors` model in Supplementary Fig. 9 appears to suggest that the actin disturbance that triggers the release of R12C and viral sensing occurs as a consequence of the penetration of the viral core through the cortical actin cytoskeleton. It seems there is something missing from this model, because the R12C-dependent sensing was dependent on proviral integration."*

As noted by the reviewer and clarified in the revised discussion (lines 336-342), actin disturbances occur early in HIV infection. However, sensing remains minimal at this stage because viral nucleic acids are shielded by intact capsids. As indicated in our model, efficient RLR-mediated sensing and its antagonism by Nef primarily occur later in the infection cycle, after proviral integration.